# LOMIA: Label-Only Membership Inference Attacks against Pre-trained Large Vision-Language Models

**Yihao Liu, Xinqi Lyu, Dong Wang, Yanjie Li, Bin Xiao**[*]
Department of Computing, The Hong Kong Polytechnic University
{yihao5.liu,xinqi.lyu,dong-comp.wang,yanjie.li}@connect.polyu.hk,
b.xiao@polyu.edu.hk

## Abstract

Large vision-language models (VLLMs) have driven significant progress in multi-modal systems, enabling a wide range of applications across domains such as healthcare, education, and content generation. Despite the success, the large-scale datasets used to train these models often contain sensitive or personally identifiable information, raising serious privacy concerns. To audit and better understand such risks, membership inference attacks (MIAs) have become a key tool. However, existing MIAs against VLLMs predominantly assume access to full-model logits, which are typically unavailable in many practical deployments. To facilitate MIAs in a more realistic and restrictive setting, we propose a novel framework: label-only membership inference attacks (LOMIA) targeting pre-trained VLLMs where only the model's top-1 prediction is available. Within this framework, we propose three effective attack methods, all of which exploit the intuition that training samples are more likely to be memorized by the VLLMs, resulting in outputs that exhibit higher semantic alignment and lower perplexity. Our experiments show that our framework surpasses existing label-only attack adaptations for different VLLMs and competes with state-of-the-art logits-based attacks across all metrics on three widely used open-source VLLMs and GPT-4o.

## 1 Introduction

Large vision-language models (VLLMs) [31, 56, 55, 36] have proven to be formidable tools capable of understanding and generating multimodal content, leading to substantial advancements in areas such as image captioning [25, 8] and visual question answering [42, 13, 28]. These models are typically trained on vast datasets that integrate images and textual descriptions, often sourced from the Internet. While this extensive training enables impressive performance, it also raises substantial privacy issues, especially in relation to the inadvertent inclusion of sensitive or proprietary information in the training data [32]. For example, identity-related details, such as names linked to faces can be revealed, even when both alignment and fine-tuning are performed using anonymized datasets [3]. In the medical domain, VLLMs trained on datasets comprising medical images and associated diagnoses may unintentionally disclose private information of patients [22, 33]. Moreover, numerous studies have demonstrated that large language models (LLMs) are capable of memorizing substantial portions of training data prior to overfitting and exhibit a lower tendency to forget during training [48, 4, 50]. Given that VLLMs are built upon LLMs, it is reasonable to expect that they inherit this memorization behavior as well.

One of the most common methods used to tackle these risks is membership inference attack (MIA), which tries to distinguish whether a particular sample has been utilized in training model [45]. However, prevailing methods largely presume access to the full set of model logits, enabling the

---

[*]Bin Xiao is the corresponding author.

39th Conference on Neural Information Processing Systems (NeurIPS 2025).

estimation of token-level probabilities. These probabilities facilitate the computation of critical statistical measures such as log-likelihood, entropy, and perplexity, which have been shown to serve as strong indicators for inferring membership status. Given this constraint, a key challenge emerges: How can we effectively infer membership when only the final output of the target model is available?

To address this problem, we propose a novel framework named LOMIA for label-only MIAs on VLLMs, which is a more realistic and challenging scenario. For any target data, LOMIA contains two stages: regression stage and inference stage. The former is trying to query the surrogate model to fit the relationship between related features and the perplexity (PPL) of generated descriptions. The latter utilizes the regression model to predict the corresponding PPL value and detect the membership by threshold.

Overall, our key contributions can be summarized as follows.

- To the best of our knowledge, we are the first to explore whether pre-trained VLLMs suffer from membership inference attacks under the label-only setting.

- We propose three straightforward yet impactful label-only MIAs targeting the pre-training phase of VLLMs. These attacks are based on text-text features, image-text features, and dual features.

- Extensive evaluations conducted on two datasets and three open-source pre-trained VLLMs demonstrate that LOMIA performs comparably to existing logits-based attacks across a range of evaluation metrics. We also show the effectiveness of our methods on the closed-source model GPT-4o, which achieved an AUC of 0.669 when evaluating the image-text feature attack method (ITFA).

## 2 Related Work

**Pre-trained Large Vision-Language Models.** Building on the success of LLMs, VLLMs integrate visual perception with language generation. Pioneering works like CLIP [39] established cross-modal alignment between images and text, laying the groundwork for deeper integration. Recent VLLMs such as LLaVA [30], MiniGPT-4 [56] and LLaMA-Adapter [12] leverage pre-trained vision encoders (e.g., CLIP-ViT [39]) and LLMs (e.g., LLaMA [49], Vicuna [7]) to fuse visual and textual tokens via lightweight projection layers. This approach allows the visual and textual components to work together effectively without retraining the entire system. With additional instruction tuning [30] on aligned image-text data, these models now perform well on a variety of multi-modal tasks, such as image captioning [25], and have become a strong foundation for many downstream applications. However, their reliance on large-scale training data raises concerns regarding memorization and unintended information leakage.

**Membership Inference Attack (MIA).** MIAs seek to determine whether a particular data sample was included in the training set of a given machine-learning model [45]. These attacks have received increasing attention due to their significant implications for data privacy, particularly when training datasets contain personally identifiable information. These attacks exploit the tendency of machine learning models to behave differently on training data compared to unseen data, primarily as a consequence of overfitting [16]. MIAs are generally categorized into three types: white-box attacks, which assume access to internal model parameters [34, 45]; grey-box attacks, which require access to model logits [27]; and black-box attacks, which rely solely on model outputs [15, 23, 45]. Black-box MIAs include simple metric-based approaches leveraging statistical signals, as well as more sophisticated techniques based on shadow models [5, 2, 23], which attempt to replicate the behavior of the target model. Extensive research has explored various MIA strategies across different machine-learning models, with notable progress in applying MIAs to classification models [45, 23], generative models [52, 9], regression models [14], and embedding models [47, 10]. Recent efforts have extended these techniques to LLMs [43, 15, 4, 51, 11]. However, relatively few studies have focused on MIA risks in multi-modal models.

Early work focuses on multi-modal classification models that combine CNN encoder (e.g., ResNet-152 [35] or VGG-16 [38]) with LSTM decoder [17]. This approach [17] assumes access to output confidences and the feasibility of constructing shadow models. Subsequent efforts extend to MIAs against CLIP [19]. In contrast, this work [19] proposes a metric-based black-box MIA using cosine

similarity and model output confidences to infer membership status. A critical limitation of prior investigations lies in their focus on smaller-scale models, which tend to overfit due to limited representational capacity. More recent studies begin to evaluate attacks on VLLMs but typically rely on access to full-model logits [27]. Such grey-box approaches compute statistical metrics like MaxRényi-K% or ModRényi* as membership signals. Nevertheless, access to token-level probabilities is rarely granted in deployed systems. To better align with real-world scenarios, Hu et al. [18] proposed a black-box membership inference attack method against fine-tuned VLLMs. However, their approach depends on shadow models and assumes access to a large amount of data highly similar to the training set, an assumption rarely met in practical deployments. To the best of our knowledge, our work is the first to explore effective label-only MIAs against pre-trained VLLMs without relying on shadow training or access to logits of the target model.

## 3 Threat model

Membership inference attack represents a class of privacy breaches aimed at machine learning models, especially those trained with sensitive information [45]. Formally, considering a trained VLLM $M_\theta$, a data sample $x$, and external adversary knowledge denoted by $K$, the definition of MIA $A$ is as follows:

$$A : x, M_\theta, K \rightarrow \{0, 1\}. \tag{1}$$

Here, 0 denotes that $x$ is not included in the training dataset of $M_\theta$, while 1 indicates that it is.

**Goal of adversary.** During the training process of a target model, data samples used for training are referred to as members, while those excluded from training are called non-members. The adversary aims to determine whether a given data sample is a member or a non-member of the training dataset.

**Adversary's Capabilities.** Given a target VLLM $\mathcal{M}_{\text{tar}}$, the adversary can only query the target model and observe its outputs, without access to its internal structure or parameters. To make the scene more realistic, the adversary is not permitted to train a shadow model or utilize any shadow dataset. Even if permitted, the computational cost of training shadow models at the scale of pre-trained VLLMs remains prohibitively high for most adversaries.

**Metrics.** We assess our method using three key metrics: area under the curve (AUC) of the receiver operating characteristic (ROC) curve, balanced accuracy, and TPR@1%FPR (true positive rate at 1% false positive rate). AUC serves as an average measure across all false positive rates. Balanced accuracy evaluates attack effectiveness by measuring prediction accuracy in a balanced dataset of members and non-members. TPR@1%FPR provides a practical metric widely adopted in prior research [5, 19].

## 4 Method

In this section, we present LOMIA, a label-only membership inference attack framework tailored for VLLMs. LOMIA comprises three complementary attack variants, each leveraging a distinct feature extraction strategy to reveal training data leakage: (1) text-text feature attack (TTFA), (2) image-text feature attack (ITFA), and (3) dual-feature attack (DUFA). All methods proposed follow a two-stage procedure consisting of regression and inference, as illustrated in Figure 1.

### 4.1 Text-Text Feature Attack (TTFA)

The core concept of TTFA is based on the VLLMs' ability to remember information from past training datasets and deliver similar responses. Consequently, an intuitive approach is to directly query the VLLMs and leverage semantic similarity between model outputs and ground truth texts to infer the membership status of training samples. However, the signal strength alone is insufficient due to potential noise in generation. To address this limitation, we incorporate the PPL of the generated description as a corrective signal, capturing confidence in generation. The joint modeling via linear regression offers a simple yet effective way to quantify the relationship between semantic similarity and generation fluency.

**Regression Stage.** In the regression stage, we establish a relationship between text similarity and PPL using a surrogate model $\mathcal{M}_{\text{sur}}$. For a target image $x_v^{(i)}$ and its corresponding ground-truth $x_t^{(i)}$

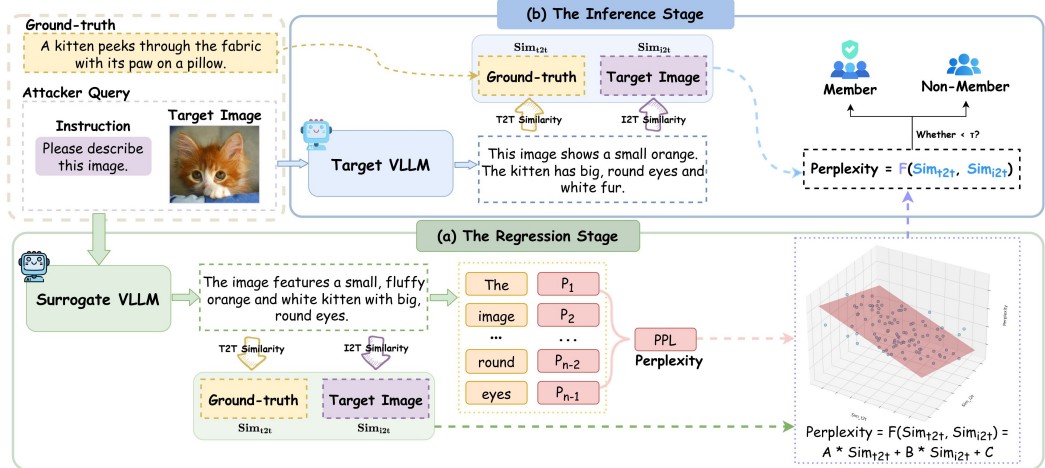

Figure 1: **Framework of LOMIA**. **(a) The Regression Stage:** We use the target samples with instructions to query the surrogate model to fit the relationship between related features and the perplexity of generated descriptions. **(b) The Inference Stage:** We first query the target model with the same target samples and instructions, then apply the regression model to predict the corresponding PPL value and detect the membership by threshold.

from the dataset $\mathcal{D}$, we use a surrogate model to generate description $\hat{x}_t^{(i)}$, and calculate the semantic similarity between generated description and ground truth $S_{\text{t2t}} = \text{sim}(\hat{x}_t^{(i)}, x_t^{(i)})$, where $\text{sim}(\cdot, \cdot)$ represents a semantic similarity function computed using a sentence embedding model. Given the generated description sequence $\hat{x}_t^{(i)} = w_1, \ldots, w_n$, we calculate the PPL of the i-th generated description:

$$PPL(\hat{x}_t^{(i)}) = \exp\left(-\frac{1}{N}\sum_{n=1}^{N}\log P(w_n|w_1, \ldots, w_{n-1})\right). \tag{2}$$

where $\log P(w_n|w_1, \ldots, w_{n-1})$ represents the logarithm of the conditional probability of the n-th word given the previous words. Finally, we establish a linear relationship between $S_{\text{t2t}}$ and $PPL$ to model their joint effect on membership inference signals, which can be formulated as follows:

$$R : S_{\text{t2t}} \mapsto PPL, via : \min_{k,b}\sum \|PPL^{(i)} - kS_{\text{t2t}}^{(i)} - b\|^2. \tag{3}$$

**Inference Stage.** During the inference stage, we apply the regression model to predict the membership score of target samples using the target model $\mathcal{M}_{\text{tar}}$. For each test sample $(x_v^{(i)}, x_t^{(i)})$, we use the target model to generate a description $\hat{x'}_t^{(i)}$ and calculate the semantic similarity between generated description and ground truth $S'_{\text{t2t}} = \text{sim}(\hat{x'}_t^{(i)}, x_t^{(i)})$. Then we apply the regression model to predict membership score and classify the sample as a member if the score exceeds a threshold $\tau$, which can be defined by:

$$\text{Score}_{\text{t2t}} = -\mathcal{R}(S'_{\text{t2t}}) = -PPL_{predicted}. \tag{4}$$

$$A(x) = \begin{cases} \text{Member}, & \text{if Score}_{\text{t2t}} \geq \tau, \\ \text{Non-member}, & \text{otherwise.} \end{cases} \tag{5}$$

## 4.2 Image-Text Feature Attack (ITFA)

ITFA is based on the hypothesis that training samples induce tighter cross-modal alignment within the learned joint embedding space. Specifically, VLLMs trained on image-text pairs tend to map member images and their generated descriptions to highly overlapping regions in the CLIP embedding space, resulting in elevated cosine similarity.

**Regression Stage.** In this stage, we establish a relationship between image-text alignment and membership signals. After querying the surrogate model $\mathcal{M}_{\text{sur}}$ to obtain the description $\hat{x}_t^{(i)}$, we utilize CLIP to calculate the cosine similarity between generated text and target image $S_{\text{i2t}} = \text{sim}(\hat{x}_t^{(i)}, x_v^{(i)})$. Next, we calculate the PPL of the generated descriptions in the same manner as TTFA and then establish a linear relationship between the image-text similarity and PPL:

$$R : S_{\text{i2t}} \mapsto PPL, via : \min_{k,b} \sum \|PPL^{(i)} - kS_{\text{i2t}}^{(i)} - b\|^2. \tag{6}$$

**Inference Stage.** The inference stage of ITFA uses the cosine similarity between text generated from the target model to predict membership score, which can be defined by:

$$\text{Score}_{\text{i2t}} = -\mathcal{R}(S'_{\text{i2t}}) = -PPL_{predicted}, where \ S'_{\text{i2t}} = \text{sim}(\hat{x'}_t^{(i)}, x_v^{(i)}). \tag{7}$$

$$A(x) = \begin{cases} \text{Member}, & \text{if Score}_{\text{i2t}} \geq \tau, \\ \text{Non-member}, & \text{otherwise.} \end{cases} \tag{8}$$

### 4.3 Dual Feature Attack (DUFA)

While TTFA and ITFA individually capture complementary aspects of memorization: semantic preservation and cross-modal alignment, they may individually fail under different sources of noise. DUFA integrates dual features in a regression model, leveraging the joint patterns in VLLMs.

**Regression Stage.** In this stage, we learn a multivariate regression model to fit both text-text and image-text similarity to the PPL. This can be formulated as follows:

$$R : (S_{\text{t2t}}, S_{\text{i2t}}) \mapsto PPL, via : \min_{k_1,k_2,b} \sum \|PPL - k_1 S_{\text{t2t}}^{(i)} - k_2 S_{\text{i2t}}^{(i)} - b\|^2. \tag{9}$$

**Inference Stage.** The inference stage of DUFA requires to use both text-text and image-text features to predict membership score, which can be defined by:

$$\text{Score}_{\text{dual}} = -\mathcal{R}(S'_{\text{t2t}}, S'_{\text{i2t}}) = -PPL_{\text{predicted}},$$
$$where \ S'_{\text{t2t}} = \text{sim}(\hat{x'}_t^{(i)}, x_t^{(i)}), S'_{\text{i2t}} = \text{sim}(\hat{x'}_t^{(i)}, x_v^{(i)}). \tag{10}$$

$$A(x) = \begin{cases} \text{Member}, & \text{if Score}_{\text{dual}} \geq \tau, \\ \text{Non-member}, & \text{otherwise.} \end{cases} \tag{11}$$

## 5 Experiments

In this section, we conduct MIAs across three target models using various baselines, and our own methods: TTFA, ITFA, and DUFA. The evaluation setup is detailed in Section 5.1. Results for TTFA, ITFA, and DUFA are presented in Section 5.2. Additionally, an ablation study is included in Section 5.3.

### 5.1 Evaluation Setup

**Models.** Our evaluation focuses on three popular open-source VLLMs: LLaVA-1.5 [30], MiniGPT-4 [56], and LLaMA-Adapter V2.1 [12], all providing full access to model weights, training process, and datasets. Here we test Vicuna-7B [7] for LLaVA and MiniGPT-4, and LLaMA-7B [49] for LLaMA-Adapter V2.1. We also test the effectiveness of our methods on closed-source model GPT-4o [1].

**Datasets.** Li et al. [27] created a dataset for MIAs on VLLMs, but it is unimodal and does not align with our task. Therefore, we developed a multi-modal dataset for MIAs against VLLMs:

- **LOMIA/LAION.** Pre-trained VLLMs such as LLaVA-1.5, MiniGPT-4, and LLaMA-Adapter V2 use images from the LAION [41], Conceptual Captions 3M [6], Conceptual 12M [6], and SBU Captions [37] datasets for pre-training [27]. Following Li et al. [27], we randomly sample a subset from the intersection of the datasets used by these three pre-trained VLLMs to serve as the member data. We then use the captions of the member data as input to query the stable-diffusion-v1-5 [40] to generate images that serve as non-member data. To ensure the validity of our MIA on VLLMs, we have 600 images in LOMIA/LAION (300 members and 300 non-members).
- **LOMIA/CC.** MS COCO [29] is also a popular dataset used in the pre-training process of the target models, so we randomly select some images in this dataset as member data. We use a similar approach to generate non-member data with stable-diffusion-v1-5 [40]. We also have 600 images in LOMIA/CC (300 members and 300 non-members).

**Baselines.** We compare our LOMIA with seven existing state-of-the-art attack methods. For logits-based baselines, we select the PPL attack [4, 54], MIN-K% PROB [44], Aug-KL [27], Max-Prob-Gap% PROB [27], MaxRényi-K% [27], and ModRényi* [27]. For label-only baselines, we mainly consider the Query Attack proposed by Hu et al. [18]. The details of the baselines can be found in the Appendix A.

**Evaluation Settings.** The attack implementation is conducted on 4 NVIDIA 3090 GPUs. To assess the performance of our LOMIA framework, we use LLaVA-1.5-7B as the surrogate model. To mitigate the impact of varying text lengths, we fix the maximum token length at 32. For computing text-text similarity, we use the all-MiniLM-L6-v2 model [21], which encodes each sentence into dense representations via mean pooling over the transformer outputs, followed by cosine similarity measurement. Image-text similarity is calculated using CLIP [39], which maps visual and textual inputs into a shared embedding space through dedicated encoders before computing their cosine similarity.

## 5.2  Main Results

We compare our methods with six advanced logits-based attack methods and one label-only attack method. Complete results on LOMIA/LAION and LOMIA/CC are summarized in Table 1, Table 2 and Table 3 respectively. Our extensive empirical evaluation has revealed several insightful observations about MIAs against pre-trained VLLMs, which we discuss in detail below.

**LOMIA performs comparably to existing logits-based attacks.** Table 1 and Table 2 show that our three attack methods based on the framework of LOMIA significantly outperforms other label-only attacks across all datasets and target model settings. Even when compared with existing logits-based attacks, our methods achieve similar or superior performance. Specifically, our TTFA method achieves the highest AUC scores among all label-only attacks on nearly all target models, consistently surpassing the baseline Query Attack by a substantial margin. For instance, TTFA improves over the Query Attack by 0.105 and 0.069 in AUC on LLaVA, as shown in Table 1 and Table 2, respectively. In terms of TPR@1%FPR, our methods remain competitive with strong logits-based baselines. For instance, TTFA achieves up to 3.0% on LLaMA Adapter shown in Table 2, clearly outperforming the Query Attack (0.0%). While top-performing logits-based methods like MaxRényi ($\alpha = 2$) still yield the highest TPR values (e.g., 5.6%–7.3%), LOMIA approaches offer a practical alternative under stricter threat models. Notably, ITFA demonstrates robust TPR gains across architectures, indicating its effectiveness even when model architectures diverge.

**Comparison among TTFA, ITFA, and DUFA.** DUFA performs best when the surrogate model shares the same architecture as the target model, since its design relies on feature-level alignment between the two (e.g., AUC = 0.621 on LOMIA/LAION and 0.630 on LOMIA/CC). However, when the surrogate and target architectures differ, DUFA's advantage diminishes, and its performance can fall below that of single-feature variants like TTFA or ITFA. This indicates that different target models vary in their sensitivity to similarity-based features, likely due to differences in architecture or training that affect their vulnerability to text-based similarity attacks in controlled settings. Notably, when we expanded the dataset and used real images, as detailed in our Appendix C, DUFA consistently outperformed the other methods across all target models, while TTFA's performance dropped significantly. This reversal demonstrates that DUFA offers greater robustness as experimental conditions become more realistic and challenging. By combining text-to-text and

image-to-text similarity features, DUFA is able to maintain strong performance across varying data scales and distributions.

Table 1: Complete results of various attacks on LOMIA/LAION.

| Metrics | AUC ↑ | | | Balanced Acc ↑ | | | TPR@1%FPR ↑ | | |
|---|---|---|---|---|---|---|---|---|---|
| Target Model | LLaVA | MiniGPT4 | LLaMA Adapter | LLaVA | MiniGPT4 | LLaMA Adapter | LLaVA | MiniGPT4 | LLaMA Adapter |
| *Logits-based Attacks* | | | | | | | | | |
| Perplexity | 0.625 | 0.597 | 0.603 | 0.611 | 0.580 | 0.586 | 3.0% | 2.6% | 2.6% |
| Aug_KL | 0.551 | 0.526 | 0.449 | 0.551 | 0.546 | 0.510 | 2.0% | 1.0% | 1.6% |
| Max-Prob-Gap | 0.588 | 0.579 | 0.562 | 0.586 | 0.565 | 0.561 | 1.3% | 2.6% | 1.3% |
| Min0% Prob | 0.676 | 0.559 | 0.549 | 0.641 | 0.551 | 0.548 | 3.3% | 1.6% | 0.3% |
| Min10% Prob | 0.738 | 0.569 | 0.564 | 0.693 | 0.553 | 0.561 | 2.0% | 2.0% | 0.6% |
| Min20% Prob | 0.661 | 0.581 | 0.573 | 0.633 | 0.573 | 0.568 | 1.6% | 1.3% | 0.6% |
| ModRényi($\alpha = 0.5$) | 0.628 | 0.588 | 0.606 | 0.601 | 0.575 | 0.583 | 4.6% | 1.6% | 1.3% |
| ModRényi($\alpha = 2$) | 0.629 | 0.587 | 0.606 | 0.601 | 0.573 | 0.581 | 2.6% | 2.6% | 1.6% |
| Max0%Rényi ($\alpha = 0.5$) | 0.616 | 0.547 | 0.629 | 0.593 | 0.548 | 0.610 | 4.6% | 3.0% | 5.3% |
| Max10%Rényi ($\alpha = 0.5$) | 0.659 | 0.559 | 0.534 | 0.620 | 0.556 | 0.538 | 4.6% | 1.3% | 2.0% |
| Max100%Rényi ($\alpha = 0.5$) | 0.581 | 0.581 | 0.563 | 0.580 | 0.575 | 0.565 | 5.3% | 1.0% | 1.0% |
| Max0%Rényi ($\alpha = 1$) | 0.679 | 0.560 | 0.618 | 0.636 | 0.560 | 0.596 | 9.6% | 3.0% | 4.3% |
| Max10%Rényi ($\alpha = 1$) | 0.750 | 0.579 | 0.549 | 0.695 | 0.568 | 0.540 | 13.0% | 3.3% | 2.6% |
| Max100%Rényi ($\alpha = 1$) | 0.625 | 0.605 | 0.589 | 0.596 | 0.588 | 0.573 | 7.3% | 1.3% | 1.6% |
| Max0%Rényi ($\alpha = 2$) | 0.696 | 0.555 | 0.576 | 0.668 | 0.556 | 0.576 | 10.0% | 3.3% | 5.6% |
| Max10%Rényi ($\alpha = 2$) | 0.775 | 0.577 | 0.565 | 0.718 | 0.560 | 0.566 | 3.6% | 2.3% | 1.3% |
| Max100%Rényi ($\alpha = 2$) | 0.631 | 0.602 | 0.599 | 0.616 | 0.581 | 0.583 | 2.3% | 1.6% | 1.6% |
| Max0%Rényi ($\alpha = \infty$) | 0.676 | 0.555 | 0.551 | 0.641 | 0.543 | 0.550 | 3.3% | 1.6% | 1.3% |
| Max10%Rényi ($\alpha = \infty$) | 0.738 | 0.569 | 0.564 | 0.693 | 0.555 | 0.561 | 2.0% | 2.0% | 1.0% |
| Max100%Rényi ($\alpha = \infty$) | 0.633 | 0.599 | 0.603 | 0.625 | 0.583 | 0.586 | 1.3% | 1.3% | 1.3% |
| *Label-only Attacks* | | | | | | | | | |
| Query Attack | 0.516 | 0.545 | 0.492 | 0.538 | 0.555 | 0.520 | 0.3% | 0.3% | 0.3% |
| TTFA (Ours) | 0.601 | **0.589** | **0.571** | **0.601** | **0.596** | **0.566** | **3.0%** | 0.6% | 2.6% |
| ITFA (Ours) | 0.617 | 0.571 | 0.522 | 0.591 | 0.568 | 0.546 | 1.6% | **2.0%** | **3.0%** |
| DUFA (Ours) | **0.621** | 0.576 | 0.526 | 0.593 | 0.563 | 0.545 | 1.6% | 1.6% | 2.3% |

**LOMIA on GPT-4o.** Table 3 demonstrates the effectiveness of our proposed attack methods within the LOMIA framework on GPT-4o. Among the three methods, ITFA achieves the best performance on the LOMIA/LAION (AUC 0.669). DUFA and TTFA exhibit consistent results across both datasets. Notably, DUFA achieves the highest AUC and Balanced Accuracy on LOMIA/CC and competitive performance on LOMIA/LAION. This indicates that even closed-source models are vulnerable to privacy attacks. Especially, LLMs like GPT-4o, which are trained on a variety of datasets, are more likely to memorize a greater amount of private information.

**Robustness across datasets.** We observe that the relative performance ranking among TTFA, ITFA, and DUFA remains largely consistent across both LOMIA/LAION and LOMIA/CC. This suggests that LOMIA is not overfitted to dataset-specific patterns and maintains attack efficiency across varying data distributions, reinforcing its practical applicability in real-world settings.

## 5.3 Ablation Study

In this section, we conduct comprehensive experiments to investigate the impact of different settings to LOMIA, including different T2T embedding models, different image-text matching models, different max token length and different temperature. Additional experimental results are available in the Appendix C.

**Different T2T embedding models.** Table 4 shows ablation experiments on LOMIA/LAION under different embedding models to explore its effectiveness, such as all-MiniLM-L6-v2 [21], bge-large-en-v1.5 [53], mxbai-embed-large-v1 [20], and UAE-Large-V1 [26]. While TTFA shows high sensitivity to embedding model choice (e.g., 3.0% TPR@1%FPR for all-MiniLM-L6-v2 but 0.6% for bge-large-en-v1.5), DUFA demonstrates remarkable robustness, maintaining AUC variations within a stable range (from 0.618 to 0.621) across various embedding models.

Table 2: Complete results of various attacks on LOMIA/CC.

| Metrics | AUC ↑ | | | Balanced Acc ↑ | | | TPR@1%FPR ↑ | | |
|---|---|---|---|---|---|---|---|---|---|
| Target Model | LLaVA | MiniGPT4 | LLaMA Adapter | LLaVA | MiniGPT4 | LLaMA Adapter | LLaVA | MiniGPT4 | LLaMA Adapter |
| *Logits-based Attacks* | | | | | | | | | |
| Perplexity | 0.634 | 0.574 | 0.556 | 0.615 | 0.573 | 0.556 | 4.6% | 3.0% | 2.6% |
| Aug_KL | 0.500 | 0.712 | 0.539 | 0.516 | 0.676 | 0.541 | 2.3% | 3.6% | 0.6% |
| Max-Prob-Gap | 0.607 | 0.615 | 0.564 | 0.601 | 0.585 | 0.560 | 2.0% | 6.3% | 2.0% |
| Min0% Prob | 0.612 | 0.775 | 0.566 | 0.600 | 0.708 | 0.558 | 3.6% | 2.0% | 0.6% |
| Min10% Prob | 0.609 | 0.880 | 0.568 | 0.603 | 0.823 | 0.565 | 5.0% | 3.0% | 1.6% |
| Min20% Prob | 0.608 | 0.825 | 0.565 | 0.588 | 0.766 | 0.560 | 4.3% | 5.6% | 1.6% |
| ModRényi($\alpha = 0.5$) | 0.633 | 0.569 | 0.549 | 0.611 | 0.581 | 0.555 | 4.0% | 1.0% | 2.0% |
| ModRényi($\alpha = 2$) | 0.633 | 0.569 | 0.544 | 0.618 | 0.580 | 0.548 | 3.6% | 1.3% | 3.0% |
| Max0%Rényi ($\alpha = 0.5$) | 0.627 | 0.808 | 0.565 | 0.598 | 0.771 | 0.560 | 2.6% | 5.0% | 2.6% |
| Max10%Rényi ($\alpha = 0.5$) | 0.646 | 0.803 | 0.566 | 0.610 | 0.798 | 0.563 | 3.0% | 2.3% | 3.6% |
| Max100%Rényi ($\alpha = 0.5$) | 0.618 | 0.569 | 0.605 | 0.590 | 0.565 | 0.596 | 1.6% | 3.3% | 1.6% |
| Max0%Rényi ($\alpha = 1$) | 0.644 | 0.814 | 0.573 | 0.610 | 0.761 | 0.561 | 2.3% | 7.3% | 2.3% |
| Max10%Rényi ($\alpha = 1$) | 0.641 | 0.847 | 0.563 | 0.598 | 0.801 | 0.570 | 3.0% | 4.3% | 3.6% |
| Max100%Rényi ($\alpha = 1$) | 0.637 | 0.643 | 0.591 | 0.611 | 0.628 | 0.585 | 2.6% | 2.1% | 3.3% |
| Max0%Rényi ($\alpha = 2$) | 0.618 | 0.799 | 0.573 | 0.590 | 0.736 | 0.563 | 4.6% | 3.3% | 1.0% |
| Max10%Rényi ($\alpha = 2$) | 0.642 | 0.891 | 0.591 | 0.615 | 0.828 | 0.573 | 6.0% | 7.0% | 2.0% |
| Max100%Rényi ($\alpha = 2$) | 0.642 | 0.711 | 0.591 | 0.621 | 0.666 | 0.575 | 4.6% | 1.6% | 4.3% |
| Max0%Rényi ($\alpha = \infty$) | 0.612 | 0.775 | 0.538 | 0.600 | 0.708 | 0.561 | 3.6% | 2.0% | 2.6% |
| Max10%Rényi ($\alpha = \infty$) | 0.610 | 0.880 | 0.585 | 0.581 | 0.823 | 0.568 | 5.0% | 3.0% | 3.6% |
| Max100%Rényi ($\alpha = \infty$) | 0.646 | 0.680 | 0.591 | 0.625 | 0.645 | 0.580 | 4.3% | 2.0% | 3.3% |
| *Label-only Attacks* | | | | | | | | | |
| Query Attack | 0.599 | 0.551 | 0.536 | 0.586 | 0.553 | 0.539 | 0.1% | 0.8% | 0.0% |
| TTFA (Ours) | 0.584 | **0.572** | 0.568 | 0.585 | **0.566** | **0.566** | 4.0% | **3.3%** | 2.3% |
| ITFA (Ours) | 0.625 | 0.540 | 0.571 | **0.605** | 0.551 | 0.560 | **4.6%** | 2.0% | 2.0% |
| DUFA (Ours) | **0.630** | 0.560 | **0.579** | 0.601 | 0.555 | 0.558 | 2.3% | 2.3% | **2.6%** |

Table 3: Performance of LOMIA on GPT-4o.

| Metrics | AUC ↑ | | Balanced Acc ↑ | | TPR@1%FPR ↑ | |
|---|---|---|---|---|---|---|
| Datasets | LAION | CC | LAION | CC | LAION | CC |
| *LOMIA* | | | | | | |
| TTFA | 0.602 | 0.600 | 0.580 | 0.588 | 2.0% | 2.0% |
| ITFA | 0.669 | 0.608 | 0.635 | 0.585 | 2.3% | 3.0% |
| DUFA | 0.612 | 0.618 | 0.585 | 0.600 | 2.0% | 2.6% |

Table 4: Performance comparison of LOMIA under different sentence transformer settings on LAION.

| Metrics | AUC ↑ | | Balanced Acc ↑ | | TPR@1%FPR ↑ | |
|---|---|---|---|---|---|---|
| Methods | TTFA | DUFA | TTFA | DUFA | TTFA | DUFA |
| *Sentence transformers* | | | | | | |
| all-MiniLM-L6-v2 | 0.601 | 0.621 | 0.601 | 0.593 | 3.0% | 1.6% |
| bge-large-en-v1.5 | 0.577 | 0.618 | 0.580 | 0.593 | 0.6% | 1.6% |
| mxbai-embed-large-v1 | 0.582 | 0.618 | 0.581 | 0.593 | 1.3% | 1.6% |
| UAE-Large-V1 | 0.584 | 0.619 | 0.581 | 0.591 | 1.3% | 1.6% |

**Different image-text matching models.** We further explore the impact of different image-text matching models on LOMIA/LAION. Including CLIP [39] and BLIP [24]. These two models differ substantially in architecture and alignment strategies: CLIP employs contrastive learning for feature alignment, while BLIP uses a generative decoder conditioned on image features. Results in Table 5 show that CLIP maintains a stable performance in both ITFA (AUC = 0.617) and DUFA (AUC = 0.621), with a balanced accuracy of 0.591-0.593, reflecting its strong robustness to single-modal noise. In contrast, BLIP fails to detect in ITFA (AUC = 0.485, balanced accuracy 0.518) but shows significant improvement in DUFA (AUC increased by 18.4% to 0.574), revealing that the additional text-text feature provides complementary information.

Table 5: Performance comparison of LOMIA under different image-text matching models settings on LAION.

| Metrics | AUC ↑ | | Balanced Acc ↑ | | TPR@1%FPR ↑ | |
| Methods | ITFA | DUFA | ITFA | DUFA | ITFA | DUFA |
|---|---|---|---|---|---|---|
| *Image-text matching models* | | | | | | |
| CLIP | 0.617 | 0.621 | 0.591 | 0.593 | 1.6% | 1.6% |
| BLIP | 0.485 | 0.574 | 0.518 | 0.585 | 0.6% | 3.0% |

**Different max token length.** We conduct ablation experiments on LOMIA/LAION targeting the length of generated description texts. we restrict the max new tokens parameter of the generation to (32, 64, 72, 96, 128). Due to CLIP's limitation of the input sequence to 77 tokens, we restrict the parameters of TTFA and DUFA to 72. Figure 2a demonstrates that longer generated description texts are not always better for TTFA and ITFA. However, for DUFA, the AUC increases as the length of the generated descriptions grows.

**Different temperature.** Since the generative behavior of VLLMs is inherently modulated by temperature settings, we performed an ablation study on LOMIA/LAION to examine the impact of temperature on attack performance. The results in Figure 2b reveal method-specific temperature sensitivities: DUFA reaches its peak at temperature = 0.4 (AUC = 0.639) and remains consistently strong within the 0.2-0.6 range (AUC > 0.62), but its performance deteriorates markedly beyond 0.7, with a 10.3% drop observed above 0.8. In contrast, ITFA is highly sensitive to temperature, performing optimally only in the low-temperature range (0.0-0.2) before rapidly degrading. TTFA follows a similar trend, yet interestingly shows a performance rebound at a temperature of 1, suggesting a potential fallback effect rather than stable robustness. Collectively, these findings underscore the necessity of modality-aware and method-specific temperature scheduling in LOMIA, preserving DUFA's operation within its stable mid-range, and leveraging the low-temperature range of TTFA and ITFA.

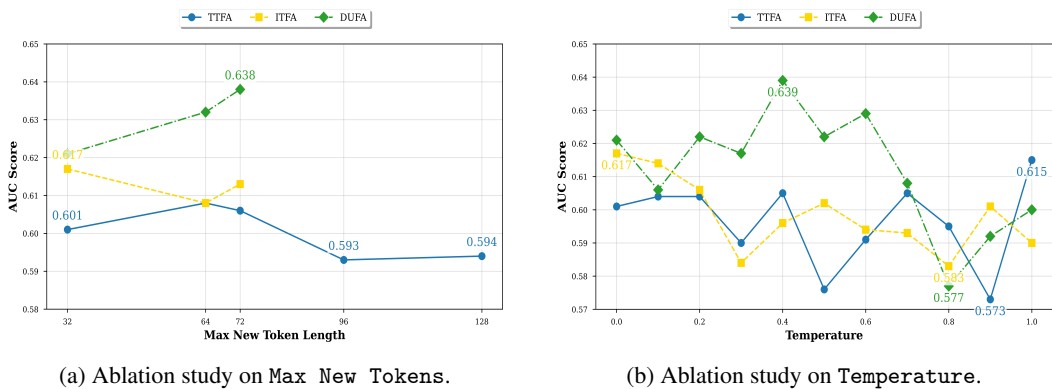

(a) Ablation study on `Max New Tokens`.      (b) Ablation study on `Temperature`.

Figure 2: **Ablation study** (a) *on different max_new_tokens.* (b) *on different temperature.*

# 6 Conclusion

In this paper, we explore the susceptibility of VLLMs to MIAs under the restrictive label-only setting. Our attention is directed towards the pre-training phase, which is not only more commonly encountered but also presents considerable technical challenges. We start by reviewing previous MIAs tailored to VLLMs and demonstrate that current methods mainly rely on logits-based attacks, which is a major limitation given the infeasibility of logits access in real-world deployments. Moreover, existing label-only approaches are inadequate when applied to pre-trained VLLMs. To address this issue, we present LOMIA, a novel label-only MIA framework that exploits sentence-level semantic similarity to approximate output perplexity as signals to distinguish membership. Comprehensive evaluations under rigorous benchmarks validate LOMIA's competitive performance over baselines. These findings highlight the urgent need for defenses specifically tailored to label-only MIAs.

## Acknowledgement

This work was supported in part by the Hong Kong Research Grants Council's (RGC) General Research Fund (GRF) under Grant PolyU 15201323.

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

# A   Details of Baselines

We summarize the details of the baselines as follows:

- **PPL Attack [4, 54]** uses the perplexity $\mathcal{PPL}$ of a target sample $x$ evaluated on the target model $M_\theta$ as the membership score, making a prediction based on whether the negative perplexity exceeds a threshold: $\mathcal{A}(x, \theta) = \mathbb{1}[-\mathcal{PPL}(x, M_\theta) \geq \tau]$.
- **Aug-KL [27]** uses KL-divergence to compare the logit distributions, where $p$ is the logit distribution of the target sample $x$ on the target model $f_\theta$, and $q$ is the logit distribution of the augmented sample: $\mathcal{A}(x, \theta) = \mathbb{1}[\mathbb{D}_{\mathrm{KL}}(p \parallel q) \geq \tau]$.
- **MIN-K% PROB [44]** determines the membership score by focusing on the k% of tokens exhibiting the lowest likelihoods: $\mathcal{A}(x, \theta) = \mathbb{1}[\frac{1}{|\mathrm{Min-K}(t)|} \sum_{t_i \in \mathrm{Min-K}(t)} \log p(t_i | t_1, \ldots, t_{i-1}) \geq \tau]$.
- **Max-Prob-Gap% PROB [27]** subtracts the second largest probability from the maximum probability in each token position and calculate the mean: $\mathcal{A}(x, \theta) = \frac{1}{|\mathrm{Max-Gap}(t)|} \sum_{t_i \in \mathrm{Max-Gap}(t)} (\max_j p_j - \mathrm{second} \max_j p_j) \geq \tau$.
- **MaxRényi-K% [27]** the top K% from the sequence $X$ with the largest Rényi entropies: $\mathcal{A}(x, \theta) = \mathbb{1}[\frac{1}{|\mathrm{Max-K\%}(X)|} \sum_{i \in \mathrm{Max-K\%}(X)} H_\alpha(p^{(i)}) \geq \tau]$.
- **ModRényi* [27]** extends `MaxRényi-K%` to the target-based scenarios: $\mathcal{A}(x, \theta) = \mathbb{1}[-\frac{1}{|\alpha-1|}\left((1-p_y)p_y^{|\alpha-1|} - (1-p_y) + \sum_{j \neq y} p_j(1-p_j)^{|\alpha-1|} - p_j\right) \geq \tau]$.
- **Query attack [18]** queries the VLLMs a number of times to generate different descriptions. Then calculate the pairwise similarity between all these descriptions and take the average value as the similarity score for the image. If the average value exceeds a certain threshold, the image is classified as a member. Here, we set the number of queries to 5 and the temperature to 0.5, which performs the best.

# B   Details of VLLMs

Table 6: VLLM details

| Model | Mini-GPT4 | LLaVA 1.5 | LLaMA Adapter v2.1 |
|---|---|---|---|
| Base LLM | Vicuna-v1.5-7B | Vicuna-v1.5-7B | LLaMA-7B |
| Vision processor | BLIP2/EVA-ViT-G | CLIP-ViT-L | CLIP-ViT-L |

# C   Additional experiments

## C.1   Performance for LOMIA on larger VLLMs

We also test our LOMIA on larger VLLMs, such as LLaVA-1.5-13b and MiniGPT4-vicuna-13b, as shown in Table 7 and Table 8.

Table 7: Performance of LOMIA on LLaVA-1.5-13b.

| Metrics | AUC ↑ | | Balanced Acc ↑ | | TPR@1%FPR ↑ | |
|---|---|---|---|---|---|---|
| Datasets | LAION | CC | LAION | CC | LAION | CC |
| *LOMIA* | | | | | | |
| TTFA | 0.601 | 0.576 | 0.596 | 0.568 | 4.0% | 4.0% |
| ITFA | 0.594 | 0.601 | 0.578 | 0.575 | 2.6% | 3.3% |
| DUFA | 0.607 | 0.609 | 0.598 | 0.581 | 3.3% | 3.3% |

## C.2   Different sample ratios on LOMIA/LAION

We explore the impact of sample ratio on the performance of LOMIA by testing ratios of 10%, 25%, 50%, 75%, and 100%. We still use the whole samples from LOMIA/LAION to fit the regression

Table 8: Performance of LOMIA on MiniGPT4-vicuna-13b.

| Metrics | AUC ↑ | | Balanced Acc ↑ | | TPR@1%FPR ↑ | |
|---|---|---|---|---|---|---|
| Datasets | LAION | CC | LAION | CC | LAION | CC |
| *LOMIA* | | | | | | |
| TTFA | 0.588 | 0.584 | 0.583 | 0.576 | 1.0% | 4.6% |
| ITFA | 0.573 | 0.539 | 0.580 | 0.546 | 0.6% | 1.0% |
| DUFA | 0.590 | 0.559 | 0.598 | 0.561 | 1.0% | 1.6% |

Table 9: Performance of TTFA under different sample settings on LOMIA/LAION.

| Ratio | AUC↑ | Balanced Acc↑ | TPR@1%FPR↑ |
|---|---|---|---|
| 10% | 0.637 | 0.662 | 6.3% |
| 25% | 0.589 | 0.604 | 1.5% |
| 50% | 0.584 | 0.589 | 3.8% |
| 75% | 0.590 | 0.594 | 3.3% |
| 100% | 0.601 | 0.601 | 3.0% |

Table 10: Performance of ITFA under different sample settings on LOMIA/LAION.

| Ratio | AUC↑ | Balanced Acc↑ | TPR@1%FPR↑ |
|---|---|---|---|
| 10% | 0.662 | 0.662 | 3.8% |
| 25% | 0.614 | 0.606 | 2.2% |
| 50% | 0.642 | 0.624 | 1.7% |
| 75% | 0.615 | 0.594 | 2.1% |
| 100% | 0.617 | 0.591 | 1.6% |

Table 11: Performance of DUFA under different sample settings on LOMIA/LAION.

| Ratio | AUC↑ | Balanced Acc↑ | TPR@1%FPR↑ |
|---|---|---|---|
| 10% | 0.589 | 0.617 | 5.3% |
| 25% | 0.614 | 0.614 | 2.1% |
| 50% | 0.612 | 0.611 | 4.4% |
| 75% | 0.600 | 0.594 | 3.5% |
| 100% | 0.621 | 0.593 | 1.6% |

model. During the inference phase, we adjust the number of samples by randomly selecting a specific ratio of samples from LOMIA/LAION to query the target model (LLaVA-1.5-7b). To ensure fairness, we average the results over five independent queries. Since the sampled data is randomly selected each time, the TTFA, ITFA, and DUFA methods are evaluated on different batches of data in each trial. Table 9, Table 10 and Table 11 show the attack results of different sample ratios.

## C.3 Balanced Accuracy Results on different Max New Tokens and on different Temperature

Figure 3a and Figure 3b show the balanced accuracy results on different max new tokens and on different temperature.

## C.4 Complete results of various attacks on larger dataset

To evaluate the impact of a larger test set, we expanded the LOMIA/LAION dataset to 1,000 samples (500 members and 500 non-members). The attack performance with this increased number of testing data points is reported in the Table 12 and Table 13.

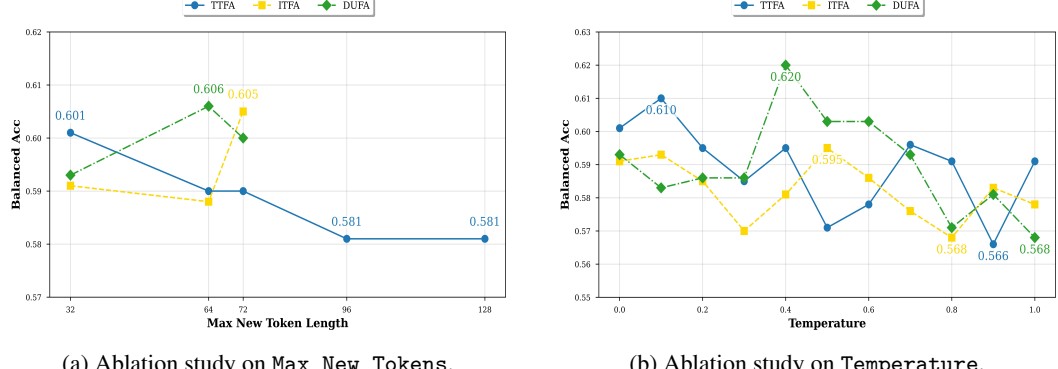

(a) Ablation study on `Max New Tokens`.  (b) Ablation study on `Temperature`.

Figure 3: **Balanced accuracy results of LOMIA on different** (a) *max_new_tokens.* (b) *temperature.*

Table 12: Complete results of various attacks on larger dataset.

| Metrics | AUC ↑ | | | Balanced Acc ↑ | | | TPR@1%FPR ↑ | | |
|---|---|---|---|---|---|---|---|---|---|
| Target Model | LLaVA | MiniGPT4 | LLaMA Adapter | LLaVA | MiniGPT4 | LLaMA Adapter | LLaVA | MiniGPT4 | LLaMA Adapter |
| *Logits-based Attacks* | | | | | | | | | |
| Perplexity | 0.628 | 0.586 | 0.588 | 0.601 | 0.570 | 0.574 | 5.2% | 2.6% | 3.6% |
| Aug_KL | 0.530 | 0.515 | 0.501 | 0.532 | 0.526 | 0.533 | 1.8% | 1.2% | 3.6% |
| Max-Prob-Gap | 0.606 | 0.574 | 0.552 | 0.591 | 0.565 | 0.557 | 1.4% | 1.4% | 0.6% |
| Min0% Prob | 0.666 | 0.565 | 0.526 | 0.641 | 0.555 | 0.532 | 3.0% | 1.6% | 1.4% |
| Min10% Prob | 0.641 | 0.563 | 0.546 | 0.659 | 0.552 | 0.546 | 4.0% | 2.6% | 1.8% |
| Min20% Prob | 0.648 | 0.571 | 0.561 | 0.618 | 0.560 | 0.562 | 4.0% | 2.8% | 2.4% |
| ModRényi($\alpha = 0.5$) | 0.629 | 0.581 | 0.588 | 0.593 | 0.568 | 0.576 | 3.0% | 2.0% | 2.0% |
| ModRényi($\alpha = 2$) | 0.630 | 0.587 | 0.589 | 0.597 | 0.573 | 0.577 | 3.0% | 2.6% | 1.6% |
| Max0%Rényi ($\alpha = 0.5$) | 0.596 | 0.528 | 0.622 | 0.574 | 0.532 | 0.594 | 3.8% | 0.6% | 2.2% |
| Max10%Rényi ($\alpha = 0.5$) | 0.664 | 0.545 | 0.547 | 0.627 | 0.547 | 0.540 | 5.2% | 1.0% | 1.2% |
| Max100%Rényi ($\alpha = 0.5$) | 0.609 | 0.569 | 0.564 | 0.590 | 0.559 | 0.560 | 5.6% | 1.4% | 1.0% |
| Max0%Rényi ($\alpha = 1$) | 0.635 | 0.559 | 0.610 | 0.594 | 0.551 | 0.591 | 5.2% | 1.2% | 4.0% |
| Max10%Rényi ($\alpha = 1$) | 0.724 | 0.571 | 0.557 | 0.666 | 0.562 | 0.550 | 7.8% | 2.4% | 1.6% |
| Max100%Rényi ($\alpha = 1$) | 0.637 | 0.589 | 0.582 | 0.604 | 0.572 | 0.569 | 6.4% | 3.0% | 1.6% |
| Max0%Rényi ($\alpha = 2$) | 0.676 | 0.565 | 0.550 | 0.645 | 0.557 | 0.546 | 5.0% | 3.2% | 3.6% |
| Max10%Rényi ($\alpha = 2$) | 0.748 | 0.573 | 0.552 | 0.688 | 0.566 | 0.562 | 4.4% | 3.0% | 2.2% |
| Max100%Rényi ($\alpha = 2$) | 0.639 | 0.590 | 0.588 | 0.612 | 0.571 | 0.570 | 5.8% | 2.8% | 1.8% |
| Max0%Rényi ($\alpha = \infty$) | 0.666 | 0.567 | 0.526 | 0.641 | 0.556 | 0.532 | 3.0% | 2.2% | 1.4% |
| Max10%Rényi ($\alpha = \infty$) | 0.704 | 0.566 | 0.545 | 0.659 | 0.551 | 0.546 | 4.0% | 2.8% | 1.8% |
| Max100%Rényi ($\alpha = \infty$) | 0.639 | 0.588 | 0.588 | 0.611 | 0.569 | 0.574 | 6.8% | 2.6% | 1.8% |
| *Label-only Attacks* | | | | | | | | | |
| Query Attack | 0.531 | 0.534 | 0.534 | 0.534 | 0.536 | 0.532 | 0.4% | 0.5% | 0.5% |
| TTFA (Ours) | 0.597 | 0.570 | 0.566 | 0.581 | 0.562 | 0.560 | **2.2%** | **2.2%** | 0.6% |
| ITFA (Ours) | 0.620 | 0.572 | 0.571 | 0.594 | 0.568 | **0.567** | 1.4% | 0.8% | **2.6%** |
| DUFA (Ours) | **0.628** | **0.576** | **0.578** | **0.603** | **0.570** | 0.564 | 2.0% | 1.0% | 2.4% |

Table 13: Performance of larger dataset on GPT-4o.

| Model | AUC ↑ | Balanced Acc ↑ | TPR@1%FPR ↑ |
|---|---|---|---|
| TTFA | 0.598 | 0.577 | 1.8% |
| ITFA | 0.648 | 0.616 | 3.2% |
| DUFA | 0.675 | 0.637 | 2.2% |

## C.5 Complete results of various attacks when non-member data are real images

To evaluate the impact of real non-member, we randomly sampled 500 image-caption pairs from open-images-captions-micro on Hugging Face [46], ensuring exclusion of any data overlapping with COCO or LAION datasets. We also enlarged our LOMIA/LAION member data samples to 500, the final results can be found in the Table 14 and Table 15.

Table 14: Complete results of various attacks when non-member data are real images.

| Metrics | AUC ↑ | | | Balanced Acc ↑ | | | TPR@1%FPR ↑ | | |
|---|---|---|---|---|---|---|---|---|---|
| Target Model | LLaVA | MiniGPT4 | LLaMA Adapter | LLaVA | MiniGPT4 | LLaMA Adapter | LLaVA | MiniGPT4 | LLaMA Adapter |
| *Logits-based Attacks* | | | | | | | | | |
| Perplexity | 0.640 | 0.589 | 0.591 | 0.604 | 0.584 | 0.573 | 3.2% | 2.8% | 2.8% |
| Aug_KL | 0.544 | 0.512 | 0.494 | 0.552 | 0.546 | 0.516 | 0.8% | 4.2% | 0.8% |
| Max-Prob-Gap | 0.620 | 0.575 | 0.567 | 0.595 | 0.570 | 0.576 | 1.6% | 2.4% | 0.8% |
| Min0% Prob | 0.592 | 0.571 | 0.521 | 0.585 | 0.565 | 0.538 | 1.0% | 1.8% | 1.0% |
| Min10% Prob | 0.538 | 0.575 | 0.533 | 0.538 | 0.569 | 0.538 | 1.2% | 2.2% | 2.0% |
| Min20% Prob | 0.571 | 0.579 | 0.588 | 0.569 | 0.572 | 0.573 | 4.0% | 1.8% | 2.8% |
| ModRényi($\alpha = 0.5$) | 0.653 | 0.583 | 0.630 | 0.623 | 0.571 | 0.598 | 3.2% | 2.4% | 6.4% |
| ModRényi($\alpha = 2$) | 0.658 | 0.588 | 0.629 | 0.627 | 0.575 | 0.593 | 3.6% | 2.4% | 5.0% |
| Max0%Rényi ($\alpha = 0.5$) | 0.555 | 0.578 | 0.629 | 0.548 | 0.571 | 0.599 | 1.2% | 1.8% | 7.2% |
| Max10%Rényi ($\alpha = 0.5$) | 0.560 | 0.570 | 0.528 | 0.553 | 0.565 | 0.537 | 2.4% | 1.4% | 1.4% |
| Max100%Rényi ($\alpha = 0.5$) | 0.611 | 0.584 | 0.547 | 0.601 | 0.578 | 0.545 | 3.6% | 2.2% | 2.2% |
| Max0%Rényi ($\alpha = 1$) | 0.601 | 0.579 | 0.654 | 0.582 | 0.565 | 0.622 | 2.2% | 1.6% | 6.8% |
| Max10%Rényi ($\alpha = 1$) | 0.710 | 0.571 | 0.574 | 0.652 | 0.562 | 0.565 | 8.2% | 2.4% | 1.6% |
| Max100%Rényi ($\alpha = 1$) | 0.642 | 0.584 | 0.580 | 0.611 | 0.579 | 0.574 | 5.8% | 2.2% | 2.2% |
| Max0%Rényi ($\alpha = 2$) | 0.596 | 0.577 | 0.693 | 0.582 | 0.564 | 0.650 | 3.4% | 1.8% | 2.8% |
| Max10%Rényi ($\alpha = 2$) | 0.688 | 0.573 | 0.586 | 0.645 | 0.566 | 0.583 | 5.6% | 3.0% | 2.6% |
| Max100%Rényi ($\alpha = 2$) | 0.623 | 0.600 | 0.602 | 0.600 | 0.637 | 0.587 | 3.6% | 3.6% | 2.6% |
| Max0%Rényi ($\alpha = \infty$) | 0.647 | 0.703 | 0.688 | 0.628 | 0.640 | 0.532 | 3.6% | 2.2% | 7.6% |
| Max10%Rényi ($\alpha = \infty$) | 0.666 | 0.710 | 0.573 | 0.634 | 0.645 | 0.572 | 3.2% | 6.2% | 3.6% |
| Max100%Rényi ($\alpha = \infty$) | 0.649 | 0.710 | 0.608 | 0.630 | 0.647 | 0.588 | 7.2% | 5.8% | 3.0% |
| *Label-only Attacks* | | | | | | | | | |
| Query Attack | 0.593 | 0.554 | 0.536 | 0.573 | 0.559 | 0.546 | 0.5% | 0.5% | 0.2% |
| TTFA (Ours) | 0.579 | 0.571 | 0.545 | 0.571 | 0.566 | 0.544 | 1.2% | **1.9%** | 1.8% |
| ITFA (Ours) | 0.630 | 0.575 | 0.575 | 0.600 | 0.569 | 0.562 | 2.2% | 1.8% | 1.6% |
| DUFA (Ours) | **0.635** | **0.580** | **0.578** | **0.605** | **0.576** | **0.566** | **2.6%** | 1.8% | **2.6%** |

Table 15: Performance of real non-member dataset on GPT-4o.

| Model | AUC ↑ | Balanced Acc ↑ | TPR@1%FPR ↑ |
|---|---|---|---|
| TTFA | 0.663 | 0.625 | 4.2% |
| ITFA | 0.609 | 0.592 | 1.2% |
| DUFA | 0.665 | 0.645 | 3.4% |

# D   Threshold Calibration

We use the AUC score, balanced accuracy, and TPR@low FPR as our primary metrics. These metrics are standard in the previous work, such as [44, 27], as both are threshold-independent. Specifically, the AUC represents the area under the ROC curve, which plots the true positive rate against the false positive rate across all possible thresholds. This means it is not necessary to select a specific threshold when comparing different MIA methods. However, if the goal is to deploy a particular MIA method in practice, a threshold can be determined by performing hyperparameter search on a small validation set.

# E   Limitations

One limitation of this work is that the TPR@1%FPR is relatively low, but it's important to note that TPR@1%FPR is an extremely stringent evaluation metric that requires high true positive rates while maintaining only 1% false positive rate. Even under this strict criterion, our method consistently outperforms existing label-only attack baselines and shows competitive performance with logits-based MIAs. Moreover, promising defense strategies include differentially private pre-training and machine unlearning, which aim to reduce memorization in VLLMs and thereby mitigate privacy risks. We leave this as a critical direction for future work.

