# OpenReview forum: "LOMIA: Label-Only Membership Inference Attacks against Pre-trained Large Vision-Language Models"
_NeurIPS.cc/2025/Conference — NeurIPS 2025 poster_

### Official Review · Reviewer_J72D · 2025-06-15

**Clarity:** 4
**Significance:** 3
**Originality:** 4
**Rating:** 4
**Confidence:** 4

**Summary:**

This paper addresses the membership inference attack (MIA) on VLLMs in the label-only setting. This paper introduces 3 attacks variants: TTFA, ITFA, DUFA. The evaluation is conducted across open-source and close-source VLLMs demonstrating competitive attack results with logits-based attacks.

**Questions:**

1. How does the attack performance change with different surrogate model? In other words, a comprehensive studies on the impact of the surrogate model to proposed attacks would comprehend this study.
2. How is the attack performance with a more number of testing data point?
3. In this paper, the author claims that this is the first study on MIAs on VLLMs. However, [17] seems to be a label-only MIA on VLLMs. Please clarify how your setting differs from [17], and if the claim of being the first still holds.

**Ethical Concerns:**

["NO or VERY MINOR ethics concerns only"]

**Final Justification:**

I have carefully reviewed authors' rebuttal.

All my concerns have been addressed.
This study is pioneering and interesting on MI attack for pre-trained VLMs.
However, my the limitation on the low TPR@1%FPR (making this less applicable) remains.
Therefore, I keep my original rating.

**Limitations:**

I suggest the authors to include a section to discuss limitations and ethical impact.

Please see my opinion on the limitations of this study in the **Strengths And Weaknesses**

**Quality:**

3

**Strengths And Weaknesses:**

**Strengths**:
- This paper tackles an important and timely problem for a realistic deployment threat model.
- The proposed attacks is simple and modular.
- The evaluations across VLLMs including GPT4-o shows competitive performance with logits-based MIAs.

**Weaknesses**:
- The attack success seems to depend on the alignment between surrogate and target models. In the realistic label-only scenario, where we might not know the underlying design (e.g., architectures) of target model, the attack performance might be degraded.
- The absolute TPR@1%FPR is low, which might not be actionable in practice with small sample batches.
- The evaluation is comprehensive in terms of number of VLLMs, but the number of testing data points is relatively small (i.e., 300/300 for membership/non-membership for each dataset)

---

> ### Author Rebuttal · Authors · 2025-07-31
>
> # Response to Reviewer J72D
>
> Thank you for your thoughtful and constructive feedback on our manuscript. We appreciate the opportunity to address your concerns and provide additional clarification. Additionally, thank you for bringing to my attention the issue with the unfilled NeurIPS reproducibility checklist. I apologize for this mistake, and I am grateful to the AC and reviewers for allowing me the chance to submit a rebuttal despite this. I will make sure to fully and accurately complete the reproducibility checklist in the camera-ready version, adhering to all NeurIPS guidelines. Furthermore, I will carefully address all the weaknesses highlighted by the reviewers in the limitations section. I appreciate your attention to this crucial aspect of responsible research reporting. Below, we respond to the key points raised.
>
> ## Response to Q1
> Q1: How does the attack performance change with different surrogate model? In other words, a comprehensive studies on the impact of the surrogate model to proposed attacks would comprehend this study.
>
> Answer: Thank you for raising this important point. We have conducted an ablation study to evaluate the impact of different surrogate models on the attack performance. The results of this comprehensive analysis are presented in the following tables.
>
> **Table: Complete results of various attacks on LOMIA/LAION when the surrogate model is LLaVA.**
>
> | Metrics / Target Model | AUC ↑ (LLaVA) | AUC ↑ (MiniGPT4) | AUC ↑ (LLaMA Adapter) | Balanced Acc ↑ (LLaVA) | Balanced Acc ↑ (MiniGPT4) | Balanced Acc ↑ (LLaMA Adapter) | TPR@1%FPR ↑ (LLaVA) | TPR@1%FPR ↑ (MiniGPT4) | TPR@1%FPR ↑ (LLaMA Adapter) |
> |-----------------------|:-------------:|:----------------:|:---------------------:|:----------------------:|:-------------------------:|:------------------------------:|:-------------------:|:----------------------:|:---------------------------:|
> | TTFA (Ours)           | 0.601         | 0.589            | 0.571                 | 0.601                  | 0.596                     | 0.566                          | 3.0%                | 0.6%                   | 2.6%                        |
> | ITFA (Ours)           | 0.617         | 0.571            | 0.522                 | 0.591                  | 0.568                     | 0.546                          | 1.6%                | 2.0%                   | 3.0%                        |
> | DUFA (Ours)           | 0.621         | 0.576            | 0.526                 | 0.593                  | 0.563                     | 0.545                          | 1.6%                | 1.6%                   | 2.3%                        |
>
> **Table: Complete results of various attacks on LOMIA/LAION when the surrogate model is MiniGPT4.**
>
> | Metrics / Target Model | AUC ↑ (LLaVA) | AUC ↑ (MiniGPT4) | AUC ↑ (LLaMA Adapter) | Balanced Acc ↑ (LLaVA) | Balanced Acc ↑ (MiniGPT4) | Balanced Acc ↑ (LLaMA Adapter) | TPR@1%FPR ↑ (LLaVA) | TPR@1%FPR ↑ (MiniGPT4) | TPR@1%FPR ↑ (LLaMA Adapter) |
> |-----------------------|:-------------:|:----------------:|:---------------------:|:----------------------:|:-------------------------:|:------------------------------:|:-------------------:|:----------------------:|:---------------------------:|
> | TTFA (Ours)           | 0.589         | 0.592            | 0.560                 | 0.575                  | 0.600                     | 0.552                          | 2.2%                | 1.6%                   | 1.0%                        |
> | ITFA (Ours)           | 0.576         | 0.583            | 0.515                 | 0.570                  | 0.583                     | 0.527                          | 1.0%                | 2.2%                   | 0.6%                        |
> | DUFA (Ours)           | 0.592         | 0.600            | 0.551                 | 0.600                  | 0.590                     | 0.563                          | 1.2%                | 1.8%                   | 1.2%                        |
>
> **Table: Complete results of various attacks on LOMIA/LAION when the surrogate model is LLaMA-Adapter.**
>
> | Metrics / Target Model | AUC ↑ (LLaVA) | AUC ↑ (MiniGPT4) | AUC ↑ (LLaMA Adapter) | Balanced Acc ↑ (LLaVA) | Balanced Acc ↑ (MiniGPT4) | Balanced Acc ↑ (LLaMA Adapter) | TPR@1%FPR ↑ (LLaVA) | TPR@1%FPR ↑ (MiniGPT4) | TPR@1%FPR ↑ (LLaMA Adapter) |
> |-----------------------|:-------------:|:----------------:|:---------------------:|:----------------------:|:-------------------------:|:------------------------------:|:-------------------:|:----------------------:|:---------------------------:|
> | TTFA (Ours)           | 0.589         | 0.586            | 0.596                 | 0.581                  | 0.588                     | 0.581                          | 2.0%                | 0.6%                   | 2.6%                        |
> | ITFA (Ours)           | 0.596         | 0.554            | 0.547                 | 0.583                  | 0.556                     | 0.550                          | 1.0%                | 1.0%                   | 2.6%                        |
> | DUFA (Ours)           | 0.600         | 0.575            | 0.605                 | 0.591                  | 0.580                     | 0.593                          | 1.6%                | 0.6%                   | 3.0%                        |
>
> ## Response to Q2
> Q2: How is the attack performance with a more number of testing data point?
>
> Answer: Thank you for raising this important point. To evaluate the impact of a larger test set, we expanded the LOMIA/LAION dataset to 1,000 samples (500 members and 500 non-members). The attack performance with this increased number of testing data points is reported in the following table （Due to character limitations, the complete comparative experiments with other methods will be presented in the revised version.）:
>
> **Table: Complete results of our attacks on larger dataset.**
>
> | Metrics / Target Model | AUC ↑ (LLaVA) | AUC ↑ (MiniGPT4) | AUC ↑ (LLaMA Adapter) | Balanced Acc ↑ (LLaVA) | Balanced Acc ↑ (MiniGPT4) | Balanced Acc ↑ (LLaMA Adapter) | TPR@1%FPR ↑ (LLaVA) | TPR@1%FPR ↑ (MiniGPT4) | TPR@1%FPR ↑ (LLaMA Adapter) |
> |-----------------------|:-------------:|:----------------:|:---------------------:|:----------------------:|:-------------------------:|:------------------------------:|:-------------------:|:----------------------:|:---------------------------:|
> | TTFA (Ours)           | 0.597         | 0.570            | 0.566                 | 0.581                  | 0.562                     | 0.560                          | 2.2%                | 2.2%                   | 0.6%                        |
> | ITFA (Ours)           | 0.620         | 0.572            | 0.571                 | 0.594                  | 0.568                     | 0.567                          | 1.4%                | 0.8%                   | 2.6%                        |
> | DUFA (Ours)           | 0.628         | 0.576            | 0.578                 | 0.603                  | 0.570                     | 0.564                          | 2.0%                | 1.0%                   | 2.4%                        |
>
> **Table: Performance of larger dataset on GPT-4o**
>
> | Model | AUC ↑ | Balanced Acc ↑ | TPR@1%FPR ↑ |
> |-------|-------|----------------|-------------|
> | TTFA  | 0.598 | 0.577          | 1.8%        |
> | ITFA  | 0.648 | 0.616          | 3.2%        |
> | DUFA  | 0.675 | 0.637          | 2.2%        |
>
> ## Response to Q3
> Q3: In this paper, the author claims that this is the first study on MIAs on VLLMs. However, [17] seems to be a label-only MIA on VLLMs. Please clarify how your setting differs from [17], and if the claim of being the first still holds.
>
> Answer: Thank you for your question. To the best of our knowledge, our work is the first to investigate label-only MIAs on pre-trained VLLMs, whereas [17] focuses on MIAs against fine-tuned VLLMs. Attacking pre-trained models is generally more challenging. This can be attributed to the fact that VLLMs are pre-trained on large-scale corpora, with each sample typically seen only once or a few times (often fewer than three). As a result, VLLMs exhibit superior generalization compared to fine-tuned VLLMs, which in turn reduces the gap in distance to the decision boundary between members and non-members. We will clarify this distinction in the revised manuscript.
>
> ## Response to W2
> W2: The absolute TPR@1%FPR is low, which might not be actionable in practice with small sample batches.
>
> Answer: Thank you for raising this important point. We agree that the absolute TPR@1%FPR are relatively low, which may limit the direct applicability of the attack in scenarios with small sample batches. However, it is worth noting that this phenomenon is not unique to our proposed method. In fact, as shown in our results and in prior work, many logit-based membership inference attacks on VLLMs also achieve similarly low or even lower TPR@1%FPR values under the same evaluation protocol. We will discuss this point in the revised manuscript.

---

> > ### Comment · Reviewer_J72D · 2025-08-05
> > **Response to authors' rebuttal**
> >
> > Thanks authors for the rebuttal.
> >
> > I strongly encourage the authors to include these additional discussion and results in the revision.
> > That said, the limitation on the low TPR@1%FPR (making this less applicable) remains.
> >
> > Therefore, I keep my original rating.

---

> > > ### Author Response · Authors · 2025-08-08
> > >
> > > We appreciate your concern regarding the TPR@1\%FPR values. However, it's important to note that TPR@1\%FPR is an extremely stringent evaluation metric that requires high true positive rates while maintaining only 1\% false positive rate. Even under this strict criterion, our method consistently outperforms existing label-only attack baselines and shows competitive performance with logits-based MIAs. We will clarify this evaluation standard and its implications in the revised manuscript.

---

### Official Review · Reviewer_HBHB · 2025-06-25

**Clarity:** 1
**Significance:** 3
**Originality:** 3
**Rating:** 4
**Confidence:** 4

**Summary:**

This paper explores a label-only membership inference attacks for pre-trained large vision-language models. Before this paper, existing MIAs against VLLM predominately assume access to full-model logits and it is unavailable in many practical deployments. Therefore, authors propose a novel framework: label-only membership inference attacks (LOMIA) targeting pre-trained VLLMs where only the model’s top-1 prediction is available. LOMIA constrcuts the relationship between sample similarities and perplexities to determine whether the sample is in member or non-member. Experiments show the effective of the proposed method.

**Questions:**

See Weaknesses.

**Ethical Concerns:**

["Major Concern: Data privacy, copyright, and consent"]

**Final Justification:**

Leaving aside the compliance of checklist, I am satisfied with this paper and the response also solves my concerns. Therefore, I increase the rating and hope the authors add more implementaion details or release the code for reproducing and pushing the development of the community.

**Limitations:**

No, this paper doesn't discuss limitations.

**Paper Formatting Concerns:**

None.

**Quality:**

3

**Strengths And Weaknesses:**

Strengths:
1. Exploring label-only membership inference attacks for pre-trained large vision-language models is helpful to the practical deployments of VLLMs.
2. The proposed method is intuitive and novel.

Weaknesses:
1. Lack of detail optimization on attacking. How to optimize the Equations 3,6 and 9?
2. Lack of the threshold of determining whether the sample is a member or non-member, which is very important to reproduce this paper.
3. The checklist is unfinished and it may conflict with the conference policy. Furthermore, this paper doesn't discuss limitations or social negative impacts.
4. Constructing the relationship between sample similarities and perplexities as an attack lacks a detailed explanation.

---

> ### Author Rebuttal · Authors · 2025-07-31
>
> # Response to Reviewer HBHB
>
> Thank you for your thoughtful and constructive feedback on our manuscript. We appreciate the opportunity to address your concerns and provide additional clarification. Below, we respond to the key points raised.
>
> ## Response to Q1
> Q1: Lack of detail optimization on attacking. How to optimize the Equations 3,6 and 9?
>
> Answer: Many thanks for your valuable suggestions. Equation 3 employs closed-form least squares optimization for text-to-text similarity mapping:
>
> $$
> R : S_{t2t} \mapsto PPL, \quad \min_{k,b} \sum \left\| PPL^{(i)} - k S_{t2t}^{(i)} - b \right\|^2
> $$
>
> where we minimize L2 loss between text-to-text similarity and PPL using the analytical solution:
>
> $$
> k = \frac{n\sum xy - \sum x \sum y}{n\sum x^2 - (\sum x)^2}
> $$
>
> $$
> b = \frac{\sum y - k\sum x}{n}
> $$
>
> with $x$ representing text-to-text similarity scores and $y$ representing PPL.
>
> Equation 6 follows the same optimization framework but leverages CLIP-based image-to-text features:
>
> $$
> R : S_{i2t} \mapsto PPL, \quad \min_{k,b} \sum \left\| PPL^{(i)} - k S_{i2t}^{(i)} - b \right\|^2
> $$
>
> utilizing identical closed-form solutions where $x$ now represents CLIP image-to-text similarity scores and $y$ representing PPL.
>
> Equation 9 extends to multivariate optimization:
>
> $$
> R : (S_{i2t}, S_{t2t}) \mapsto PPL, \quad \min_{k_1, k_2, b} \sum \left\| PPL - k_1 S_{t2t}^{(i)} - k_2 S_{i2t}^{(i)} - b \right\|^2
> $$
>
> employing normal equations:
>
> $$
> \boldsymbol{\beta} = (\mathbf{X}^T \mathbf{X})^{-1} \mathbf{X}^T \mathbf{y}
> $$
>
> where $\boldsymbol{\beta} = [k_1, k_2, b]^T$, feature matrix $\mathbf{X}$ combines both text-to-text and image-to-text similarities, and $y$ representing PPL.
>
> ## Response to Q2
> Q2: Lack of the threshold of determining whether the sample is a member or non-member, which is very important to reproduce this paper.
>
> Answer: Thank you for raising this important point. In our evaluation, we use the AUC score, balanced accuracy, and TPR@low FPR as our primary metrics. These metrics are standard in the previous work, such as [3], as both are threshold-independent. Specifically, the AUC represents the area under the ROC curve, which plots the true positive rate against the false positive rate across all possible thresholds. This means it is not necessary to select a specific threshold when comparing different MIA methods. However, if the goal is to deploy a particular MIA method in practice, a threshold can be determined by performing hyperparameter search on a small validation set. We will clarify this explanation in the revised version to enhance reproducibility.
>
> ## Response to Q3
> Q3: The checklist is unfinished and it may conflict with the conference policy. Furthermore, this paper doesn't discuss limitations or social negative impacts.
>
> Answer: Thank you for pointing out the omission regarding the NeurIPS reproducibility checklist. I sincerely apologize for this oversight and appreciate the AC and reviewers’ understanding in still providing the opportunity for rebuttal. I will ensure that the checklist is properly completed in the camera-ready version, in full compliance with NeurIPS requirements. Additionally, I'll consider all the weaknesses provided to include in our limitations section. Thank you again for your attention to this important aspect of responsible reporting.
>
> ## Response to Q4
> Q4: Constructing the relationship between sample similarities and perplexities as an attack lacks a detailed explanation.
>
> Answer: Thank you for your valuable question. The detailed explanation for the relationship between sample similarities and perplexities are as follows:
> First, it is well established that LLMs tend to assign higher probabilities to tokens and sequences that are more semantically or syntactically similar to their training data. This means that, for member samples, the model is more likely to generate outputs with higher confidence, resulting in lower perplexity scores. Conversely, for non-member samples, the model is less familiar with the content, leading to higher perplexity. Second, the process of text generation in LLMs inherently favors high-probability tokens, further amplifying the difference in perplexity between member and non-member samples. This property is inherited by VLLMs, as they are built upon LLM architectures and share similar probabilistic behaviors. To provide a deeper analysis, we also examine the relationship between relevant features and perplexity. Our results, summarized using standard regression metrics in the following table.
>
> **Table: Standard regression metrics for regression models**
>
> | Method | TTFA   | ITFA   | DUFA   |
> |--------|--------|--------|--------|
> | R²     | 0.4074 | 0.3174 | 0.4824 |
> | RMSE   | 0.0358 | 0.0384 | 0.0335 |
>
> ## References
>
> [3] Ko M, Jin M, Wang C, et al. "Practical membership inference attacks against large-scale multi-modal models: A pilot study." ICCV 2023.

---

> > ### Comment · Reviewer_HBHB · 2025-08-07
> >
> > Leaving aside the compliance of checklist, I am satisfied with this paper and the response also solves my concerns. Therefore, I increase the rating and hope the authors to add more implementaion details or release the code for reproducing and pushing the development of the community.

---

> > > ### Author Response · Authors · 2025-08-08
> > >
> > > We sincerely thank the reviewer HBHB, for the detailed review of our paper, as well as the recognition and recommendation! We will add more implementation details or release the code in the revised version.

---

### Official Review · Reviewer_gVAv · 2025-06-30

**Clarity:** 3
**Significance:** 2
**Originality:** 3
**Rating:** 4
**Confidence:** 4

**Summary:**

This paper presents LOMIA, a novel label-only membership inference attack (MIA) framework targeting pre-trained large vision-language models (VLLMs). Unlike prior works that assume access to logits or use shadow models, LOMIA operates in a realistic black-box setting where only the model’s top-1 output is observable. The key idea is to use the semantic similarity between generated outputs and ground-truth captions as input to a surrogate regression model that predicts the perplexity (PPL) of the generated text. Membership is inferred based on whether the predicted PPL falls below a fixed threshold. The authors propose three attack variants (TTFA, ITFA, DUFA) and demonstrate their effectiveness across multiple open-source and closed-source VLLMs, including GPT-4o.

**Questions:**

### **Questions**

1. **How do attacks perform when real unseen images (e.g., held-out from COCO/CC or external datasets) are used as non-members instead of diffusion-generated ones?**
   Diffusion-generated images may introduce domain shifts. Evaluating the attack with real non-members would help isolate the impact of memorization vs. unfamiliarity in visual appearance.

2. **Have the authors explored the sensitivity of the attack to the choice of threshold τ?**
   Given that PPL scales may differ across models, has percentile-based or target-specific calibration been considered to improve generalization?

3. **Can the authors report standard regression metrics (e.g., RMSE, R²) for the sim→PPL model?**
   Since the regression model underlies the final attack decision, it would be helpful to quantify its accuracy and reliability directly.

4. **Could the authors clearly specify the vision encoder used in each target model (LLaVA, MiniGPT-4, LLaMA-Adapter)?**
   The paper mentions “e.g., CLIP-ViT”, but it is unclear whether all models—especially MiniGPT-4—used CLIP-based encoders. Clarifying this is important for understanding the attack’s generality.

5. **More broadly, could the authors address the various concerns raised under the Weaknesses section of this review?**
   In particular, it would be helpful to see additional discussion or evidence regarding:
   - The generality of the attack beyond CLIP-based models (e.g., InstructBLIP)
   - How performance is affected by differences in model size (e.g., 7B vs. 13B)
   - The potential bias introduced by the sentence embedding model

   Responses to these points would strengthen the case for the method’s robustness and broader applicability.

**Ethical Concerns:**

["NO or VERY MINOR ethics concerns only"]

**Final Justification:**

I maintain my score of 4 (Borderline Accept). The paper addresses a realistic and underexplored threat model for label-only membership inference attacks against VLLMs and presents a conceptually clean and practically applicable methodology. The rebuttal was thorough and addressed many of my concerns, particularly regarding evaluation with real images, threshold calibration, regression accuracy reporting, and vision encoder specification.

**Limitations:**

The paper does not include an explicit "Limitations" section, nor does it adequately discuss key threats to validity. Specifically, the similarity between surrogate and target models is not critically examined, the use of CLIP-based similarity features is not evaluated on non-CLIP architectures, and the reliance on diffusion-generated non-members may introduce domain shift. These factors may affect the generality and robustness of the attack but are not acknowledged. Including a dedicated Limitations section would strengthen the transparency and credibility of the work.

**Paper Formatting Concerns:**

**The NeurIPS reproducibility checklist at the end of the submission appears to be left unfilled, with all items marked as To do**. This goes against the NeurIPS submission requirements and raises concerns about the authors' attention to reproducibility and responsible reporting. The authors are encouraged to complete the checklist properly in the camera-ready version.

**Quality:**

2

**Strengths And Weaknesses:**

### **Strengths**

1. **Realistic and underexplored threat model**
   The paper addresses a practically relevant but underexplored attack setting: performing membership inference on *pre-trained* VLLMs under a strict **label-only black-box** condition where only the top-1 prediction is observable. This setup closely reflects real-world deployment environments, including closed-source systems like GPT-4o, making the problem formulation highly impactful.

2. **Clever and modular methodology**
   The use of a surrogate regression model to estimate PPL from semantic similarity (text-text and image-text) is conceptually clean and effective. This sim→PPL mapping cleverly bypasses the need for logits or internal model access and enables membership inference based on a single top-1 output, making the method compatible with real-world constraints.

3. **Black-box applicability and zero-shot transfer**
   The paper successfully applies its attack to GPT-4o without access to internal structure, logits, or retraining, showing that the attack generalizes across architectures. This reinforces the practicality and transferability of the approach from open-source to commercial models.

4. **Competitive performance under strict assumptions**
   Despite the lack of logits or shadow data, LOMIA achieves performance comparable to state-of-the-art logits-based attacks across multiple VLLMs. This supports the effectiveness of the similarity-based regression framework in capturing membership signals.

---

### **Weaknesses**

1. **Use of diffusion-generated negatives may introduce domain shift**
   Non-member samples are generated using Stable Diffusion from captions. These images may differ stylistically or semantically from natural images in the training data. As a result, the model may flag them as non-members not because of lack of memorization, but due to unfamiliar visual features, conflating domain shift with membership status.

2. **Lack of principled threshold calibration**
   The attack decision relies on whether the predicted PPL falls below a fixed threshold τ. However, τ is globally fixed and not adapted to each target model, despite clear differences in perplexity scales across architectures. This could lead to misclassification, especially when transferring across models. No analysis of threshold sensitivity or calibration strategy is provided.

3. **Potential bias from embedding models**
   The sentence embedding models (e.g., MiniLM) used to compute semantic similarity may share domain overlap with the training data (e.g., LAION, CC3M), leading to inflated similarity scores regardless of memorization. The paper does not control for or discuss this possible confounder, which could skew the attack's reliability.

4. **No evaluation of regression accuracy**
   Although the regression model (sim → PPL) is central to the approach, no standard regression metrics (e.g., MAE, RMSE, R²) are reported. This makes it hard to assess how well the surrogate regression model performs, or how regression noise propagates to final attack decisions.

5. **Potential overreliance on architecturally similar surrogates**
   Although the paper presents the method as shadow-free, the surrogate and target models used in the experiments (e.g., LLaVA, MiniGPT-4, LLaMA-Adapter) may be architecturally similar. The paper does not explicitly specify the vision encoders used in each model, including MiniGPT-4, which is known to use a non-CLIP encoder (BLIP-2 ViT-G/14) by default. If CLIP-based encoders were used across all models, the results might reflect strong alignment with CLIP-specific representations. The lack of experiments on structurally different VLLMs such as InstructBLIP, BLIP-2, or ViT-G/Q-former-based architectures limits our ability to assess the generality and robustness of the attack design across vision backbones.

---

> ### Author Rebuttal · Authors · 2025-07-31
>
> # Response to Reviewer gVAv
>
> Thank you for your thoughtful and constructive feedback on our manuscript. We appreciate the opportunity to address your concerns and provide additional clarification. Below, we respond to the key points raised.
>
> ## Response to Q1
> Q1: How do attacks perform when real unseen images (e.g., held-out from COCO/CC or external datasets) are used as non-members instead of diffusion-generated ones?
>
> Answer: Many thanks for your suggestions. We randomly sampled 500 image-caption pairs from open-images-captions-micro (2024 release) on Hugging Face, ensuring exclusion of any data overlapping with COCO or LAION datasets. We also enlarge our LOMIA/LAION member data samples to 500 (500 members + 500 non-members), the final results can be found in the following table. （Due to character limitations, the complete comparative experiments with other methods will be presented in the revised version.）
>
> **Table: Complete results of our attacks when non-member data are real images**
>
> | Metrics / Target Model | AUC ↑ (LLaVA) | AUC ↑ (MiniGPT4) | AUC ↑ (LLaMA Adapter) | Balanced Acc ↑ (LLaVA) | Balanced Acc ↑ (MiniGPT4) | Balanced Acc ↑ (LLaMA Adapter) | TPR@1%FPR ↑ (LLaVA) | TPR@1%FPR ↑ (MiniGPT4) | TPR@1%FPR ↑ (LLaMA Adapter) |
> |-----------------------|:-------------:|:----------------:|:---------------------:|:----------------------:|:-------------------------:|:------------------------------:|:-------------------:|:----------------------:|:---------------------------:|
> | TTFA (Ours)           | 0.579         | 0.571            | 0.545                 | 0.571                  | 0.566                     | 0.544                          | 1.2%                | 1.9%                   | 1.8%                        |
> | ITFA (Ours)           | 0.630         | 0.575            | 0.575                 | 0.600                  | 0.569                     | 0.562                          | 2.2%                | 1.8%                   | 1.6%                        |
> | DUFA (Ours)           | 0.635         | 0.580            | 0.578                 | 0.605                  | 0.576                     | 0.566                          | 2.6%                | 1.8%                   | 2.6%                        |
>
> **Table: Performance of real non-member dataset on GPT-4o**
>
> | Model | AUC ↑ | Balanced Acc ↑ | TPR@1%FPR ↑ |
> |-------|-------|----------------|-------------|
> | TTFA  | 0.663 | 0.625          | 4.2%        |
> | ITFA  | 0.609 | 0.592          | 1.2%        |
> | DUFA  | 0.665 | 0.645          | 3.4%        |
>
> ## Response to Q2
> Q2: Have the authors explored the sensitivity of the attack to the choice of threshold τ?
>
> Answer: Thank you for raising this important point. In our evaluation, we use the AUC score, balanced accuracy, and TPR@low FPR as our primary metrics. These metrics are standard in the previous work, such as [1][3], as both are threshold-independent. Specifically, the AUC represents the area under the ROC curve, which plots the true positive rate against the false positive rate across all possible thresholds. This means it is not necessary to select a specific threshold when comparing different MIA methods. However, if the goal is to deploy a particular MIA method in practice, a threshold can be determined by performing hyperparameter search on a small validation set. We will clarify this explanation in the revised version to enhance reproducibility.
>
> ## Response to Q3
> Q3: Can the authors report standard regression metrics (e.g., RMSE, R²) for the sim→PPL model?
>
> Answer: Many thanks for your suggestions. The standard regression metrics for regression models are listed in the following Table.
>
> **Table: Standard regression metrics for regression models**
>
> | Method | TTFA   | ITFA   | DUFA   |
> |--------|--------|--------|--------|
> | R²     | 0.4074 | 0.3174 | 0.4824 |
> | RMSE   | 0.0358 | 0.0384 | 0.0335 |
>
> ## Response to Q4
> Q4: Could the authors clearly specify the vision encoder used in each target model (LLaVA, MiniGPT-4, LLaMA-Adapter)?
>
> Answer: Many thanks for your suggestions. The versions and base models of target VLLMs are listed in the following Table.
>
> **Table: VLLM details**
>
> | Model                | Mini-GPT4      | LLaVA 1.5      | LLaMA Adapter v2.1 |
> |----------------------|----------------|----------------|--------------------|
> | Base LLM             | Vicuna-v1.5-7B | Vicuna-v1.5-7B | LLaMA-7B           |
> | Vision processor     | BLIP2/EVA-ViT-G| CLIP-ViT-L     | CLIP-ViT-L         |
>
> ## Response to Q5
> Q5: More broadly, could the authors address the various concerns raised under the Weaknesses section of this review?
>
> **W1: The generality of the attack beyond CLIP-based models.**
> Answer: We have evaluated study when surrogate model is Mini-GPT4, the vision processor for Mini-GPT4 is BLIP2/EVA-ViT-G, the results are listed in the following table:
>
> **Table: Complete results of our attacks on LOMIA/LAION when the surrogate model is MiniGPT4**
>
> | Metrics / Target Model | AUC ↑ (LLaVA) | AUC ↑ (MiniGPT4) | AUC ↑ (LLaMA Adapter) | Balanced Acc ↑ (LLaVA) | Balanced Acc ↑ (MiniGPT4) | Balanced Acc ↑ (LLaMA Adapter) | TPR@1%FPR ↑ (LLaVA) | TPR@1%FPR ↑ (MiniGPT4) | TPR@1%FPR ↑ (LLaMA Adapter) |
> |-----------------------|:-------------:|:----------------:|:---------------------:|:----------------------:|:-------------------------:|:------------------------------:|:-------------------:|:----------------------:|:---------------------------:|
> | TTFA (Ours)           | 0.589         | 0.592            | 0.560                 | 0.575                  | 0.600                     | 0.552                          | 2.2%                | 1.6%                   | 1.0%                        |
> | ITFA (Ours)           | 0.576         | 0.583            | 0.515                 | 0.570                  | 0.583                     | 0.527                          | 1.0%                | 2.2%                   | 0.6%                        |
> | DUFA (Ours)           | 0.592         | 0.600            | 0.551                 | 0.600                  | 0.590                     | 0.563                          | 1.2%                | 1.8%                   | 1.2%                        |
>
> **W2: How performance is affected by differences in model size (e.g., 7B vs. 13B)**
>
> Answer: We also test our LOMIA on larger VLLMs, such as LLaVA-1.5-13b and MiniGPT4-vicuna-13b, which can be found in appendix:
>
> **Table: Performance of LOMIA on LLaVA-1.5-13b**
>
> | Metrics / Datasets | AUC ↑ (LAION) | AUC ↑ (CC) | Balanced Acc ↑ (LAION) | Balanced Acc ↑ (CC) | TPR@1%FPR ↑ (LAION) | TPR@1%FPR ↑ (CC) |
> |--------------------|:-------------:|:----------:|:----------------------:|:-------------------:|:-------------------:|:----------------:|
> | TTFA               | 0.601         | 0.576      | 0.596                  | 0.568               | 4.0%                | 4.0%             |
> | ITFA               | 0.594         | 0.601      | 0.578                  | 0.575               | 2.6%                | 3.3%             |
> | DUFA               | 0.607         | 0.609      | 0.598                  | 0.581               | 3.3%                | 3.3%             |
>
> **Table: Performance of LOMIA on MiniGPT4-vicuna-13b**
>
> | Metrics / Datasets | AUC ↑ (LAION) | AUC ↑ (CC) | Balanced Acc ↑ (LAION) | Balanced Acc ↑ (CC) | TPR@1%FPR ↑ (LAION) | TPR@1%FPR ↑ (CC) |
> |--------------------|:-------------:|:----------:|:----------------------:|:-------------------:|:-------------------:|:----------------:|
> | TTFA               | 0.588         | 0.584      | 0.583                  | 0.576               | 1.0%                | 4.6%             |
> | ITFA               | 0.573         | 0.539      | 0.580                  | 0.546               | 0.6%                | 1.0%             |
> | DUFA               | 0.590         | 0.559      | 0.598                  | 0.561               | 1.0%                | 1.6%             |
>
> **W3: The potential bias introduced by the sentence embedding model**
>
> Answer: Thank you for raising this important point. We acknowledge that some sentence embedding models (e.g., MiniLM) may have been partially trained on datasets such as COCO or LAION, which are also used in VLLM training. However, the sentence embedding model is used solely for measuring semantic similarity, not for text generation. Since embedding models encode text into vector representations to capture semantic relationships rather than generate content. Even if the embedding model was trained on datasets that overlap with VLLM training data, this would not introduce bias in our similarity measurements because: (1) the embedding model's role is to map semantically similar texts to nearby points in vector space, which is a fundamentally different task from text generation; (2) our evaluation focuses on whether the VLLM can generate captions that are semantically similar to ground truth, which requires the embedding model to have good general semantic understanding rather than memorization of specific samples. We will clarify this point and discuss the potential impact in the revised manuscript.
>
> ## Reference
>
> [1] Shi, Weijia, et al. "Detecting Pretraining Data from Large Language Models." ICLR 2024.
>
> [2] He Y, Li B, Liu L, et al. "Towards label-only membership inference attack against pre-trained large language models." USENIX Security 2025.
>
> [3] Ko M, Jin M, Wang C, et al. "Practical membership inference attacks against large-scale multi-modal models: A pilot study." ICCV 2023.

---

> > ### Comment · Reviewer_gVAv · 2025-08-04
> > **Follow-up Questions for Authors**
> >
> > Thank you for your detailed and thoughtful rebuttal — we truly appreciate the effort to address reviewers' concerns across multiple dimensions.
> >
> > We would like to follow up with two questions:
> >
> > 1. **Is there any binary classification experiment (in your paper or prior work) that directly evaluates whether similarity scores (sim) alone are sufficient for membership inference?**
> >    If not, we would appreciate a clear explanation of *why sim alone is not used directly as a membership signal* in your framework. Since this is central to understanding your attack design, a convincing justification would help readers understand the added value of regressing to PPL.
> >
> > 2. **Regarding the table in Response to Q1 (using real images as non-members)** — what should we compare these results against?
> >    As this setup uses real images rather than diffusion-generated ones, we are unsure how to interpret or benchmark these results.
> >    With only this table, it is difficult to assess how LOMIA performs on natural images *relative to other methods*. A clarification or contextual note in the revision would be helpful to guide reviewers.
> >
> > Thank you again for your thorough rebuttal!

---

> > > ### Author Response · Authors · 2025-08-08
> > >
> > > Thanks for your further comments. We will clarify the concerns.
> > > ## Response to additional Q1
> > > Thank you for this insightful question. While similarity scores can indeed serve as a membership signal, they have inherent limitations when used alone. The similarity score primarily measures the surface-level semantic overlap between the generated and reference captions, which can be affected by paraphrasing or stylistic variations. In contrast, the PPL reflects the model’s internal confidence and familiarity with a given caption, capturing more information.
> > > By regressing the similarity score to PPL, we effectively translate the external similarity signal into the model’s internal probability space, which is more sensitive to subtle differences between member and non-member samples. Prior work on MIA against LLMs has explored using similarity scores to approximate the probability of each token, then calculating sentence perplexity as a membership signal[1]. This work demonstrates that regressing to perplexity yields more robust and reliable results. This supports our design choice: although similarity alone can be informative, leveraging the predicted perplexity provides a stronger membership signal.
> > > ## Response to additional Q2
> > > Thank you for pointing out the need for clearer benchmarking. In our experiments, the set of member samples remains the same; we only change the non-member samples from diffusion-generated images to real images. This setup allows for a direct comparison of LOMIA’s performance under different non-member distributions. By holding the member set constant, any change in attack performance can be attributed to the difference in the non-member data.
> > > ## Reference
> > > [1] He Y, Li B, Liu L, et al. "Towards label-only membership inference attack against pre-trained large language models." USENIX Security 2025.

---

> > > > ### Comment · Area_Chair_97Ai · 2025-08-08
> > > >
> > > > Dear Reviewer,
> > > >
> > > > Could you please reply to the author's response and potentially update your rating? Thank you.
> > > >
> > > > Best,
> > > > AC

---

### Official Review · Reviewer_WrZE · 2025-07-01

**Clarity:** 2
**Significance:** 2
**Originality:** 2
**Rating:** 4
**Confidence:** 3

**Summary:**

This paper proposes LOMIA, a label-only Membership Inference Attack (MIA) targeting Vision-Language Models (VLMs), where the attacker has access only to the final output and no other internal model information or parameters. LOMIA leverages a surrogate model to learn the relationship between related features and Perplexity (PPL). During the inference stage, it uses this learned relationship to estimate the PPL value and determine whether a given input sample was part of the model's training data

**Questions:**

1) How is the threshold value determined for deciding whether an input is a member or non-member?


2) Can the authors provide more details about the surrogate models and clarify whether their training data differs from that of the target model?


3) Can the authors provide results on more recent VLMs?

4) Can the authors provide a clear justification or explanation for why the proposed method is effective? A deeper analysis or theoretical insight would help strengthen the contribution.

**Ethical Concerns:**

["NO or VERY MINOR ethics concerns only"]

**Final Justification:**

Thanks authors for the rebuttal. Your response addresses most of my concerns.

**Limitations:**

The authors did not include the limitation or potential negative societal impact of their work.

**Quality:**

2

**Strengths And Weaknesses:**

**Strengths**
- The paper addresses a challenging and realistic threat model, where the attacker has access only to limited information (i.e., final output labels).

- Experimental results demonstrate the effectiveness of the proposed method under this constrained setting.

**Weaknesses**

- The motivation behind the proposed approach is not clearly described. The method relies on a surrogate model to learn the relationship between certain features and PPL. This relationship is then used to estimate the PPL values of outputs from the target model. If the estimated PPL exceeds a predefined threshold, the input is classified as a member; otherwise, it is considered a non-member.
 However, this method makes a strong and potentially unrealistic assumption: that there exists a meaningful and transferable relationship between the surrogate model and the target model. For example, if the surrogate model is trained on natural image-text pairs and the target model is trained on domain-specific data such as medical images, there is no guarantee that the surrogate's regression parameters will generalize or offer meaningful guidance for the target model.


- The evaluation uses 300 member and 300 non-member samples. However, the two sets differ in nature—member samples are real data, whereas non-member samples are synthetic images generated by Stable Diffusion v1.5. This discrepancy introduces a distributional gap, which may make it easier for the model to distinguish non-member samples, thus inflating the attack performance.


- Surrogate models are assumed not to have seen the target training data. However, the paper lacks sufficient evidence or details to support this claim. Without clear verification, the results are questionable.


- The target models evaluated in the paper were released around 2023, which may be outdated given the rapid advancement in VLMs. The authors are encouraged to evaluate their method on more recent models, such as those released in late 2024 or early 2025, to strengthen the relevance and robustness of their findings.

---

> ### Author Rebuttal · Authors · 2025-07-31
>
> # Response to Reviewer WrZE
>
> Thank you for your thoughtful and constructive feedback on our manuscript. We appreciate the opportunity to address your concerns and provide additional clarification. Below, we respond to the key points raised.
>
> ## Response to Q1
> Q1: How is the threshold value determined for deciding whether an input is a member or non-member?
>
> Answer: Thank you for raising this important point. In our evaluation, we use the AUC score, balanced accuracy, and TPR@low FPR as our primary metrics. These metrics are standard in the previous work, such as [1][3], as both are threshold-independent. Specifically, the AUC represents the area under the ROC curve, which plots the true positive rate against the false positive rate across all possible thresholds. This means it is not necessary to select a specific threshold when comparing different MIA methods. However, if the goal is to deploy a particular MIA method in practice, a threshold can be determined by performing hyperparameter search on a small validation set. We will clarify this explanation in the revised version to enhance reproducibility.
>
> ## Response to Q2
> Q2: Can the authors provide more details about the surrogate models and clarify whether their training data differs from that of the target model?
>
> Answer: Thank you for your question. In this paper, we use LLaVA-1.5-7B as the surrogate model, the based LLM for LLaVA-1.5-7B is Vicuna v1.5 7B and the vision processor is CLIP-ViT-L. To ensure the validity of the experiment, we randomly sampled a subset from the intersection of the datasets used by the pre-trained target model mentioned in our paper. The use of a surrogate model is solely aimed at identifying the relationship between relevant features and the perplexity of generated descriptions, rather than determining whether a specific sample is a member. Additionally, [2] similarly exhibits training data overlap across surrogate models and target models, with no observed effect on experimental conclusions.
>
> ## Response to Q3
> Q3: Can the authors provide results on more recent VLMs?
>
> Answer: Thank you for your suggestion. As shown in Table 3, we have evaluated our methods on GPT-4o, which is one of the most recent and widely used VLLMs. This evaluation demonstrates that the privacy issues identified in our study are also present in state-of-the-art VLLMs. The detailed results can be found in the following table.
>
> **Table: Performance of LOMIA on GPT-4o**
>
> | Metrics / Datasets | AUC ↑ (LAION) | AUC ↑ (CC) | Balanced Acc ↑ (LAION) | Balanced Acc ↑ (CC) | TPR@1%FPR ↑ (LAION) | TPR@1%FPR ↑ (CC) |
> |--------------------|:-------------:|:----------:|:----------------------:|:-------------------:|:-------------------:|:----------------:|
> | **LOMIA**          |               |            |                        |                     |                     |                  |
> | TTFA               | 0.602         | 0.600      | 0.580                  | 0.588               | 2.0%                | 2.0%             |
> | ITFA               | 0.669         | 0.608      | 0.635                  | 0.585               | 2.3%                | 3.0%             |
> | DUFA               | 0.612         | 0.618      | 0.585                  | 0.600               | 2.0%                | 2.6%             |
>
> ## Response to Q4
> Q4: Can the authors provide a clear justification or explanation for why the proposed method is effective? A deeper analysis or theoretical insight would help strengthen the contribution.
>
> Answer: Thank you for your valuable question. The effectiveness of our proposed method can be attributed to the following reasons:
> First, it is well established that LLMs tend to assign higher probabilities to tokens and sequences that are more semantically or syntactically similar to their training data. This means that, for member samples, the model is more likely to generate outputs with higher confidence, resulting in lower perplexity scores. Conversely, for non-member samples, the model is less familiar with the content, leading to higher perplexity. Second, the process of text generation in LLMs inherently favors high-probability tokens, further amplifying the difference in perplexity between member and non-member samples. This property is inherited by VLLMs, as they are built upon LLM architectures and share similar probabilistic behaviors. To provide a deeper analysis, we also examine the relationship between relevant features and perplexity. Our results, summarized using standard regression metrics in the following table.
>
> **Table: Standard regression metrics for regression models**
>
> | Method | TTFA   | ITFA   | DUFA   |
> |--------|--------|--------|--------|
> | R²     | 0.4074 | 0.3174 | 0.4824 |
> | RMSE   | 0.0358 | 0.0384 | 0.0335 |
>
> ## Response to W3
> Many thanks for your suggestions. To evaluate our attacks on real unseen images, we randomly sampled 500 image-caption pairs from open-images-captions-micro (2024 release) on Hugging Face, ensuring exclusion of any data overlapping with COCO or LAION datasets. We also enlarged our LOMIA/LAION member data samples to 500 (500 members + 500 non-members), the final results can be found in the following table（Due to character limitations, the complete comparative experiments with other methods will be presented in the revised version.）:
>
> **Table: Complete results of various attacks when non-member data are real images**
>
> | Metrics / Target Model | AUC ↑ (LLaVA) | AUC ↑ (MiniGPT4) | AUC ↑ (LLaMA Adapter) | Balanced Acc ↑ (LLaVA) | Balanced Acc ↑ (MiniGPT4) | Balanced Acc ↑ (LLaMA Adapter) | TPR@1%FPR ↑ (LLaVA) | TPR@1%FPR ↑ (MiniGPT4) | TPR@1%FPR ↑ (LLaMA Adapter) |
> |-----------------------|:-------------:|:----------------:|:---------------------:|:----------------------:|:-------------------------:|:------------------------------:|:-------------------:|:----------------------:|:---------------------------:|
> | TTFA (Ours)           | 0.579         | 0.571            | 0.545                 | 0.571                  | 0.566                     | 0.544                          | 1.2%                | 1.9%                   | 1.8%                        |
> | ITFA (Ours)           | 0.630         | 0.575            | 0.575                 | 0.600                  | 0.569                     | 0.562                          | 2.2%                | 1.8%                   | 1.6%                        |
> | DUFA (Ours)           | 0.635         | 0.580            | 0.578                 | 0.605                  | 0.576                     | 0.566                          | 2.6%                | 1.8%                   | 2.6%                        |
>
> **Table: Performance of real non-member dataset on GPT-4o**
>
> | Model | AUC ↑ | Balanced Acc ↑ | TPR@1%FPR ↑ |
> |-------|-------|----------------|-------------|
> | TTFA  | 0.663 | 0.625          | 4.2%        |
> | ITFA  | 0.609 | 0.592          | 1.2%        |
> | DUFA  | 0.665 | 0.645          | 3.4%        |
>
> ## Reference
>
> [1] Shi, Weijia, et al. "Detecting Pretraining Data from Large Language Models." ICLR 2024.
>
> [2] He Y, Li B, Liu L, et al. "Towards label-only membership inference attack against pre-trained large language models." USENIX Security 2025.
>
> [3] Ko M, Jin M, Wang C, et al. "Practical membership inference attacks against large-scale multi-modal models: A pilot study." ICCV 2023.

---

> ### Comment · Reviewer_WrZE · 2025-08-06
>
> Thanks the authors for providing the rebuttal.
>
> > We also enlarged our LOMIA/LAION member data samples to 500 (500 members + 500 non-members),
>
> The updated results are interesting. Compared to Table 1, the performance rankings appear reversed. In Table 1, TTFA appears to be the most effective attack method. However, in this updated setting with real images, DUFA generally performs best, while TTFA consistently ranks the lowest. Could the authors clarify/explain this?
>
> > Response to Q4
>
> Thank you for the explanation. However, it does not fully address my concern. As I pointed out in my review, a key weakness of the method lies in its strong and potentially unrealistic, assumption that a meaningful and transferable relationship exists between the surrogate and target models. For instance, if the surrogate model is trained on natural image-text pairs, while the target model is trained on domain-specific data (e.g., medical images), there is no guarantee that the surrogate will generalize or provide useful the relationship between relevant features and the perplexity of generated descriptions.
>
>
> > Response to Q2
>
> I think it is essential that the training data for surrogate models differ from that of the target model. This is important because the paper focuses on **label-only** MIA. **If there is overlap between the training data of surrogate and target models, then the label-only setting becomes less meaningful, as access through the surrogate model undermines the assumption of label-only restriction.**

---

> > ### Author Response · Authors · 2025-08-08
> >
> > Thanks for your further comments. We will clarify the concerns.
> > ## Response to W3
> > Thank you for this insightful observation. The changes in performance rankings reveal important aspects of feature robustness under different experimental conditions. In the original Table 1, TTFA achieved the highest AUC on MiniGPT4 (0.589) and LLaMA Adapter (0.571), while DUFA performed best only on LLaVA (0.621). This indicates that different target models vary in their sensitivity to similarity-based features, likely due to differences in architecture or training that affect their vulnerability to text-based similarity attacks in controlled settings. However, when we expanded the dataset and used real images, DUFA consistently outperformed the other methods across all target models (LLaVA: 0.635, MiniGPT4: 0.580, LLaMA Adapter: 0.578), while TTFA’s performance dropped significantly. This reversal demonstrates that DUFA offers greater robustness as experimental conditions become more realistic and challenging. By combining text-to-text and image-to-text similarity features, DUFA is able to maintain strong performance across varying data scales and distributions.
> > ## Response to Q2 \& Q4
> > We appreciate your concern about training data overlap and would like to clarify our methodology and its validity. First, our approach learns a feature-to-perplexity mapping from the surrogate model, which captures general behavioral patterns of VLLMs rather than memorizing specific training samples. This mapping is then applied to extract membership signals from the regression model's perplexity responses, making the actual training data content less critical than the learned feature relationships.
> > Regarding domain transferability, while we acknowledge that extreme domain shifts (e.g., medical images) may reduce effectiveness. However, our experiments demonstrate robust transferability across different model architectures (LLaVA, MiniGPT4, LLaMA Adapter, GPT-4o) that have varying training procedures and data compositions. The key insight is that our method exploits fundamental properties of how VLLMs process multimodal information, which are relatively consistent across models trained on similar modalities, regardless of specific dataset overlap. Importantly, the label-only restriction remains meaningful because the attacker cannot directly access the target model's training data and must rely entirely on the target model's responses to infer membership, with the surrogate model only providing a regression process to predict the PPL values.

---

> > > ### Comment · Area_Chair_97Ai · 2025-08-08
> > >
> > > Dear Reviewer,
> > >
> > > Could you please reply to the author's response and potentially update your rating? Thank you.
> > >
> > > Best,
> > > AC

---

> > > ### Comment · Reviewer_WrZE · 2025-08-08
> > >
> > > Thanks the authors for the response.
> > >
> > > > The changes in performance rankings reveal important aspects of feature robustness under different experimental conditions. In the original Table 1, TTFA achieved the highest AUC on MiniGPT4 (0.589) and LLaMA Adapter (0.571), while DUFA performed best only on LLaVA (0.621). This indicates that different target models vary in their sensitivity to similarity-based features, likely due to differences in architecture or training that affect their vulnerability to text-based similarity attacks in controlled settings. However, when we expanded the dataset and used real images, DUFA consistently outperformed the other methods across all target models (LLaVA: 0.635, MiniGPT4: 0.580, LLaMA Adapter: 0.578), while TTFA’s performance dropped significantly. This reversal demonstrates that DUFA offers greater robustness as experimental conditions become more realistic and challenging. By combining text-to-text and image-to-text similarity features, DUFA is able to maintain strong performance across varying data scales and distributions.
> > >
> > > With these new results, I suggest updating Section 5.2, as the results, observations, and explanations appear to contradict the actual findings, particularly in the comparison among TTFA, ITFA, and DUFA.
> > >
> > > I will update my rating after discussing with the other reviewers.

---

### Decision · Program_Chairs · 2025-09-17

**Decision:**

Accept (poster)

**Comment:**

This paper presents a novel label-only membership inference attack (MIA) framework targeting pre-trained large vision-language models (VLLMs). Initially, the reviewers had concerns on the motivation, evaluation, experimental setting, etc. During the rebuttal, the authors successfully persuaded all reviewers to provide postive ratings at the end, with only minor concerns. The AC has to mention that the checklist of the paper must be carefully prepared in order to avoid desk rejection. The AC recommends acceptance of the paper, and asks the authors to finish the checklist part seriously.